# Staged and Physics-Grounded Learning Framework with Hyperintensity Prior for Pre-Contrast MRI Synthesis

**Dayang Wang** [* 1] **Srivathsa Pasumarthi** [* 1] **Ajit Shankaranarayanan** [1] **Greg Zaharchuk** [2]

## Abstract

Contrast-enhanced MRI enhances pathological visualization but often necessitates Pre-Contrast images for accurate quantitative analysis and comparative assessment. However, Pre-Contrast images are frequently unavailable due to time, cost, or safety constraints, or they may suffer from degradation, making alignment challenging. This limitation hinders clinical diagnostics and the performance of tools requiring combined image types. To address this challenge, we propose a novel staged, physics-grounded learning framework with a hyperintensity prior to synthesize Pre-Contrast images directly from Post-Contrast MRIs. The proposed method can generate high-quality Pre-Contrast images, thus, enabling comprehensive diagnostics while reducing the need for additional imaging sessions, costs, and patient risks. To the best of our knowledge, this is the first Pre-Contrast synthesis model capable of generating images that may be interchangeably used with standard-of-care Pre-Contrast images. Extensive evaluations across multiple datasets, sites, anatomies, and downstream tasks demonstrate the model's robustness and clinical applicability, positioning it as a valuable tool for contrast-enhanced MRI workflows.

## 1. Introduction

Contrast enhanced Magnetic Resonance Imaging (MRI) is a critical tool in clinical diagnostics. It is widely used to highlight pathological regions such as tumors, lesions, and vascular abnormalities. While Post-Contrast images provide

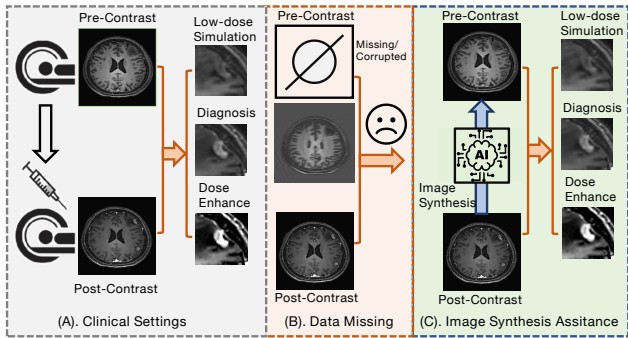

*Figure 1.* (A). Utilization of both Pre-Contrast and Post-Contrast images in clinical workflows. (B). Challenges arise when Pre-Contrast images are missing or significantly corrupted. (C). Synthesized images enable the successful completion of these tasks.

enhanced visualization of specific structures, Pre-Contrast images are often required for comparative assessment, baseline evaluation, and quantitative analysis. For example, Pre-Contrast image is used as an inevitable reference for the brain tumor imaging starting from diagnosis, through therapy planning, to treatment response and/or recurrence assessment (Villanueva-Meyer et al., 2017; Martucci et al., 2023). Zheng et. al. proposed a 4D deep learning model leveraging Pre-Contrast, arterial, and portal venous phases of dynamic contrast enhanced MRI to improve hepatocellular carcinoma lesion segmentation (Zheng et al., 2022). In clinical practice, Pre- and Post-Contrast images are typical used in a combined fashion for real settings such as low dose simulations (Sourbron & Buckley, 2011; Wang et al., 2023), diagnosis (Villanueva-Meyer et al., 2017; Zheng et al., 2022), dose enhancement (Bône et al., 2022; Subedi et al., 2012; OCHI et al., 2014; Pasumarthi et al., 2021), etc. as shown in Fig. 1(A).

However, as shown in Fig. 1(B), Pre-Contrast images are often unavailable or severely degraded due to various factors, including motion, aliasing, and streaking artifacts caused by patient movement or technical issues during data acquisition (Zaitsev et al., 2015; Graves & Mitchell, 2013). These challenges hinder the reliability and quality of imaging, which can subsequently affect diagnostic accuracy and clinical

*Equal contribution [1]Subtle Medical, 883 Santa Cruz Ave Suite 205, Menlo Park, CA, 94025 [2]Department of Neuroradiology, Stanford University, 1201 Welch Rd, MC 5488, Stanford, CA, 94305. Correspondence to: Dayang Wang <dayang@subtlemedical.com>, Srivathsa Pasumarthi <sri@subtlemedical.com>.

*Proceedings of the 42nd International Conference on Machine Learning*, Vancouver, Canada. PMLR 267, 2025. Copyright 2025 by the author(s).

decision-making (Cui et al., 2023). Addressing this limitation is critical, especially for advanced imaging modalities where Pre-Contrast images are integral to procedures like quantitative parameter mapping or contrast agent enhancement analysis.

To address this issue, a natural thought is to synthesize Pre-Contrast images directly from Post-Contrast images. Nevertheless, directly training a simple image-to-image synthesis network (e.g. UNet (Ronneberger et al., 2015)) to map Post-Contrast to Pre-Contrast images often fails to balance the synthesis of the structural and contrast information in the image. Contrast agent uptake introduces non-linear changes in tissue intensity, which are further influenced by variations in relaxation times ($\mathbf{T}_1$ and $\mathbf{T}_2$) and anatomical structure (Sourbron & Buckley, 2011). As a result, naive methods tend to produce blurry or unrealistic reconstructions that lack the fine-grained details necessary for clinical decision-making. Due to these difficulties, progress in this field has been limited, Xue et. al. proposed the first bidirectional Post-to-Pre Contrast image synthesis model with an information disentanglement design (Xue et al., 2022). However, their methods generate very blurring and distorted contrast details, making it unuseful for clinical applications. To date, no learning framework has demonstrated the ability to consistently synthesize Pre-Contrast images with the high quality and structural accuracy required for clinical applications.

As a unique solution to this problem, we propose the first MRI physics guided framework capable of synthesizing Pre-Contrast MRI images with exceptionally high quality. Our approach integrates a brightness-aware segmentation module, leveraging contrast agent uptake dynamics, with an advanced inpainting+[1] learning design for accurate Pre-Contrast reconstruction. The segmentation module employs soft-thresholding to isolate contrast-enhanced regions, guided by the relationship between uptake and tissue intensity, and constrains the inpainting+ process to ensure realistic synthesis. By explicitly modeling contrast enhancement dynamics, our method addresses the limitations of direct mappings and achieves an unprecedented level of performance in Pre-Contrast image synthesis. The contribution this paper is summarized as follows:

- We present a significant advancement in MRI image synthesis by developing a method capable of generating high-quality Pre-Contrast images to address clinical scenarios involving missing/corrupted data, demonstrating the value of AI for science.

- We identify the reason for limited progress in this area with direct image synthesis and propose a staged and physics-grounded design, coupled with a staged learn-

---
[1] '+' denotes its difference from the traditional inpainting task.

ing framework that incorporates hyperintensity priors, paving the way for future advancements.

- We extend our model with a variant capable of leveraging information from corrupted images, further enhancing synthesis results.

- The proposed methods are extensively evaluated on two real-world datasets we collected from two hospitals, demonstrating their robustness.

- We conduct comprehensive downstream experiments to validate the utility of the synthesized images for clinical applications.

## 2. Methods

In this paper, we propose a Staged and PHysics-grounded learning framework with hypErintensity prioR for Pre-Contrast MRI synthEsis (SPHERE). The proposed model features a novel dual-staged learning including uptake segmentation and inpainting processes. Our proposed method is well-grounded in MRI physics and is theorically supported. This two-stage design ensures a physics-informed and structured methodology for synthesizing Pre-Contrast MRI with hyperintensity priors.

### 2.1. Mathematical Derivation for Pre-Contrast MRI Synthesis

#### 2.1.1. MODEL DERIVATION

The contrast-enhanced magnetic resonance imaging (CE-MRI) is commonly performed with spoiled gradient-recalled echo (SPGR) sequence. The signal intensity of the Post-Contrast condition at steady state is characterised as (Bernstein, 2004):

$$\mathbf{S}_{\text{post}} = \rho \sin(\alpha) \frac{1 - \mathbf{E}_{\text{post}}}{1 - \cos(\alpha)\mathbf{E}_{\text{post}}}, \quad (1)$$

$$\mathbf{E}_{\text{post}} = e^{-\frac{\text{TR}}{\mathbf{T}1_{\text{post}}}}, \quad (2)$$

where $\mathbf{T}1_{\text{post}}$ is Longitudinal relaxation time after contrast; $\mathbf{S}_{\text{pre}}$ and $\mathbf{S}_{\text{post}}$ are the Pre-Contrast and Post-Contrast signal intensities, respectively; $\rho$ is the proton density that we assume a constant value (Kwong et al., 1992); $\alpha$ is the flip angle of the sequence.

Solving for Eq. 1, we have

$$\mathbf{E}_{\text{post}} = \frac{\mathbf{S}_{\text{post}} - \rho \sin(\alpha)}{\mathbf{S}_{\text{post}} \cos(\alpha) - \rho \sin(\alpha)}. \quad (3)$$

Similarly, for a desired Pre-Contrast MRI unknown, we have

$$\mathbf{S}_{\text{pre}} = \rho \sin(\alpha) \frac{1 - \mathbf{E}_{\text{pre}}}{1 - \cos(\alpha)\mathbf{E}_{\text{pre}}}, \quad (4)$$

$$\mathbf{E}_{\text{pre}} = e^{-\frac{\text{TR}}{\text{T1}_{\text{pre}}}}, \tag{5}$$

$$\mathbf{E}_{\text{pre}} = \frac{\mathbf{S}_{\text{pre}} - \rho \sin(\alpha)}{\mathbf{S}_{\text{pre}} \cos(\alpha) - \rho \sin(\alpha)}, \tag{6}$$

where $\mathbf{S}_{\text{pre}}$ represent the signal intensity of the Pre-Contrast image. $\mathbf{T1}_{\text{pre}}$ is Longitudinal relaxation time before contrast.

The relationship between the longitudinal relaxation times before and after contrast agent administration is given by (Hathout & Jamshidi, 2012):

$$\frac{1}{\mathbf{T1}_{\text{post}}} = \frac{1}{\mathbf{T1}_{\text{pre}}} + r_1[\text{Gd}], \tag{7}$$

Where $r_1$ is Longitudinal relaxivity of the specific contrast agent (related to T1 shortening effect). [Gd] is the concentration of the gadolinium contrast agent. Then, we have

$$\mathbf{E}_{\text{post}} = \mathbf{E}_{\text{pre}} \cdot e^{-\text{TR} \cdot r_1[\text{Gd}]}. \tag{8}$$

Given that the exact concentration of gadolinium-based contrast agent ([Gd]) is not explicitly known, we introduce an uptake mask, denoted as $\mathbf{M}$, which is a learnable variable in our model. Here, $\mathbf{M} \in [0, 1]$ represents the degree of contrast uptake within the tissue. Assuming that the contrast agent concentration is proportional to this uptake mask, i.e., $[\text{Gd}] \propto \mathbf{M}$, the equation can be reformulated as follows:

$$\mathbf{E}_{\text{post}} = \mathbf{E}_{\text{pre}} \cdot e^{-\text{TR} \cdot r_1 \cdot \beta \cdot \mathbf{M}}, \tag{9}$$

where $\beta$ is scaling factor. Substituting the derived expressions for $\mathbf{E}_{\text{pre}}$ and $\mathbf{E}_{\text{post}}$:

$$\frac{\mathbf{S}_{\text{post}} - \rho \sin(\alpha)}{\mathbf{S}_{\text{post}} \cos(\alpha) - \rho \sin(\alpha)} = \left( \frac{\mathbf{S}_{\text{pre}} - \rho \sin(\alpha)}{\mathbf{S}_{\text{pre}} \cos(\alpha) - \rho \sin(\alpha)} \right) e^{-\text{TR} \cdot r_1 \beta \mathbf{M}}. \tag{10}$$

Rearranging the equation to solve for $\mathbf{S}_{\text{pre}}$ in terms of $\mathbf{S}_{\text{post}}$:

$$\mathbf{S}_{\text{pre}} = \frac{\rho \sin(\alpha)\left(\mathbf{S}_{\text{post}} e^{\text{TR} r_1 \beta \mathbf{M}} - \mathbf{S}_{\text{post}} \cos(\alpha) - \rho e^{\text{TR} r_1 \beta \mathbf{M}} \sin(\alpha) + \rho \sin(\alpha)\right)}{\mathbf{S}_{\text{post}} e^{\text{TR} r_1 \beta \mathbf{M}} \cos(\alpha) - \mathbf{S}_{\text{post}} \cos(\alpha) - \frac{\rho e^{\text{TR} r_1 \beta \mathbf{M}} \sin(2\alpha)}{2} + \rho \sin(\alpha)}. \tag{11}$$

Since only $\mathbf{S}_{\text{post}}$ is available as input, we model the uptake mask $\mathbf{M}$ as a function of $\mathbf{S}_{\text{post}}$. As a result, directly learning $\mathbf{S}_{\text{pre}}$ from $\mathbf{S}_{\text{post}}$ using an image-to-image model becomes intractable due to the composition of functions. To address this, we propose decomposing the synthesis process into two cascaded stages, following the formulation derived above:

$$\mathbf{S}_{\text{pre}} = f_{\text{inpaint+}}(\mathbf{S}_{\text{post}}, \mathbf{M}), \quad \mathbf{M} = f_{\text{seg}}(\mathbf{S}_{\text{post}}). \tag{12}$$

The first stage involves learning the uptake mask $\mathbf{M}$ using $f_{\text{seg}}$, formulated as a segmentation task. In the second stage, $f_{\text{inpaint+}}$ utilizes $\mathbf{S}_{\text{post}}$ and $\mathbf{M}$ to compute Eq. 11, effectively

reconstructing $\mathbf{S}_{\text{pre}}$. We define this as a new 'inpainting+' task, where the '+' signifies its distinction from traditional image inpainting. Unlike standard inpainting, which primarily focuses on restoring missing regions, this approach leverages contextual information from the masked area, ensuring a more accurate reconstruction of Pre-Contrast images.

### 2.1.2. COMPLEXITY ANALYSIS

In typical direct learning method, we aim to approximate the mapping:

$$\mathbf{S}_{\text{pre}} = f_{\text{direct}}(\mathbf{S}_{\text{post}}). \tag{13}$$

However, based on the physical model, $\mathbf{S}_{\text{pre}}$ depends both directly on $\mathbf{S}_{\text{post}}$ and indirectly through an intermediate variable $\mathbf{M}$:

$$\mathbf{S}_{\text{pre}} = f_{\text{inpaint+}}\big(\mathbf{S}_{\text{post}}, f_{\text{seg}}(\mathbf{S}_{\text{post}})\big). \tag{14}$$

The segmentation function $f_{\text{seg}}$ introduces nonlinear spatial dependencies, which require a complex deep learning model. The reconstruction function $f_{\text{inpaint+}}$ introduces physical modeling dependencies, involving exponential and trigonometric terms. The network must optimize both segmentation and reconstruction in one go, which multiplies the complexity.

This formulation results in a composite function, introducing cascading dependencies. By applying the chain rule for the derivative of $\mathbf{S}_{\text{pre}}$ with respect to $\mathbf{S}_{\text{post}}$:

$$\frac{d\mathbf{S}_{\text{pre}}}{d\mathbf{S}_{\text{post}}} = \frac{\partial \mathbf{S}_{\text{pre}}}{\partial \mathbf{S}_{\text{post}}} + \frac{\partial \mathbf{S}_{\text{pre}}}{\partial \mathbf{M}} \cdot \frac{d\mathbf{M}}{d\mathbf{S}_{\text{post}}}. \tag{15}$$

The derivative indicates that any small error in $\mathbf{M}$ propagates into the reconstruction function, making optimization challenging. This coupled dependency results in a multiplicative complexity scaling, expressed as:

$$\mathcal{C}_{\text{direct}} = \mathcal{O}(C_{\text{inpaint+}} \cdot C_{\text{seg}}) \tag{16}$$

However, in our two-step learning approach, the problem is split into separate tasks of $f_{\text{seg}}$ and $f_{\text{inpaint+}}$. Each function is learned independently, and the complexity of the total process is:

$$\mathcal{C}_{\text{two-stage}} = \mathcal{O}(C_{\text{inpaint+}}) + \mathcal{O}(C_{\text{seg}}) \ll \mathcal{O}(C_{\text{inpaint+}} \cdot C_{\text{seg}}). \tag{17}$$

In Appendix. C, we also provided a comparative analysis of our method against traditional learning approaches from a loss optimization perspective.

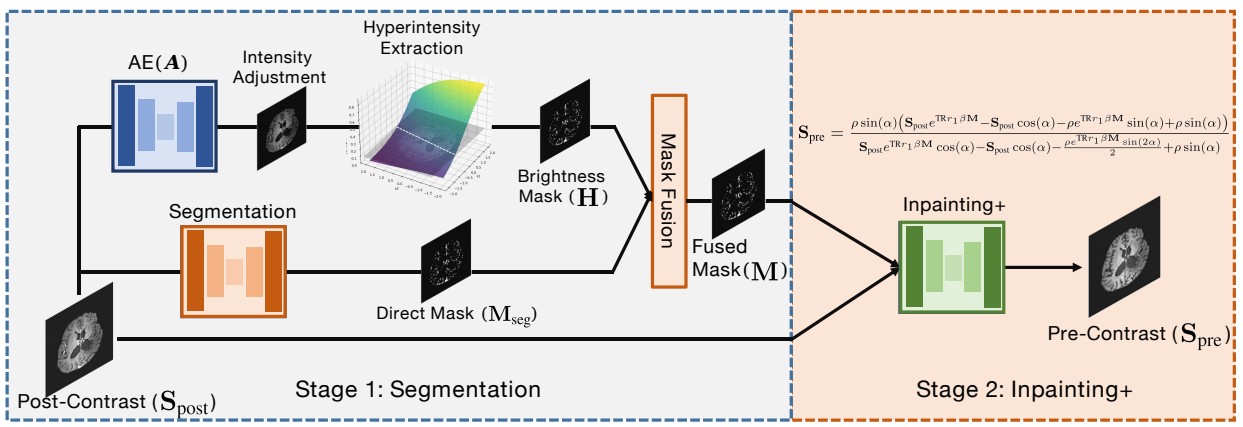

*Figure 2.* The architecture of the dual-staged learning framework

## 2.2. Decoupled Two-Stage Learning Framework

### 2.2.1. STAGE 1: LEARNING THE CONTRAST MASK

The uptake mask $\mathbf{M}$ learning is formulated as a segmentation task aimed at identifying regions with contrast enhancement. The segmentation model $f_{\text{seg}}$ is designed to predict $\mathbf{M}$ directly from $\mathbf{S}_{\text{post}}$:

$$\mathbf{M} = f_{\text{seg}}(\mathbf{S}_{\text{post}}; \theta_{\text{seg}}), \tag{18}$$

where $\theta_{\text{seg}}$ represents the learnable parameters of the segmentation model.

Specifically, the segmentation process is divided into two stages: (1) a conventional deep segmentation model to guarantee model precision and robustness (e.g. NestedUNet (Zhou et al., 2018)), and (2) A hyperintensity-prior-guided refinement module to enhance the segmentation by focusing on regions exhibiting hyperintense signals typically associated with contrast enhancement.

***Hyperintensity Prior*** To incorporate the physical constraint that contrast uptake regions exhibit higher intensity values in Post-Contrast MRI, we introduce a brightness prior to guide the derivation of the mask $\mathbf{M}$. This mask represents the likelihood of contrast agent uptake at each pixel and is constructed using a soft thresholding mechanism informed by the normalized brightness of the Post-Contrast image.

To enhance hyperintensity extraction, $\mathbf{S}_{\text{post}}$ is first processed through an autoencoder $A(\cdot)$, which adjusts image intensity for improved contrast representation. The output is then scaled by a learnable factor $\gamma$, resulting in:

$$\mathbf{S}'_{\text{post}} = \gamma A(\mathbf{S}_{\text{post}}), \tag{19}$$

where normalization ensures consistency across varying image brightness ranges. The processed intensity $\mathbf{S}'_{\text{post}}$ is then mapped to the hyperintensity mask $\mathbf{H}$ using a Softmax function $\sigma(\cdot)$:

$$\mathbf{H} = \sigma(u(\mathbf{S}'_{\text{post}} - \beta_0)), \tag{20}$$

where $u$ controls the sensitivity of the soft threshold. $\beta_0$ serves as the brightness threshold, ensuring that high-intensity regions, indicative of contrast uptake, receive higher probabilities. With $\gamma$ and $u$ as learnable parameters, the model dynamically adjusts, reducing dependence on a predefined $\beta_0$. This adaptive thresholding enhances robustness across contrast agents and anatomies, improving generalization while minimizing manual tuning.

The final segmentation map $\mathbf{M}$ is obtained by combining the outputs of the standard segmentation model $\mathbf{M}_{\text{seg}}$ and the hyperintensity prior $\mathbf{H}$ through a fusion operation:

$$\mathbf{M} = \text{Fuse}(\mathbf{M}_{\text{seg}}, \mathbf{H}), \tag{21}$$

where $\text{Fuse}(\cdot)$ is a $1 \times 1$ Convolution operation to combine the two masks into one.

The true uptake mask $\mathbf{M}_{\text{true}}$, representing regions of contrast enhancement, is approximated by the subtraction imaging technique (Hubbard et al., 2019):

$$\mathbf{M}_{\text{true}} = \mathbb{I}(\mathbf{S}_{\text{post}} - \mathbf{S}_{\text{pre}} > \tau), \tag{22}$$

where $\mathbb{I}(\cdot)$ is the indicator function that outputs 1 if the condition is true and 0 otherwise. $\tau$ is the brightness threshold to remove the background noise. The *Binary Cross Entropy* (BCE) loss is employed for model training. To further regularize this relationship, a penalty term

$$\mathcal{R}_{\text{hyper}} = \sum |\mathbf{M} - \sigma(u(\mathbf{S}'_{\text{post}} - \beta))|^2 \tag{23}$$

is introduced, enforcing adherence to the prior.

### 2.2.2. STAGE 2: LEARNING THE INPAINTING+ PROCESS

Using the predicted mask $\mathbf{M}$ and $\mathbf{S}_{\text{post}}$, the inpainting+ model $f_{\text{inpaint+}}$ synthesizes the $\mathbf{S}_{\text{pre}}$ which is defined in Eq.

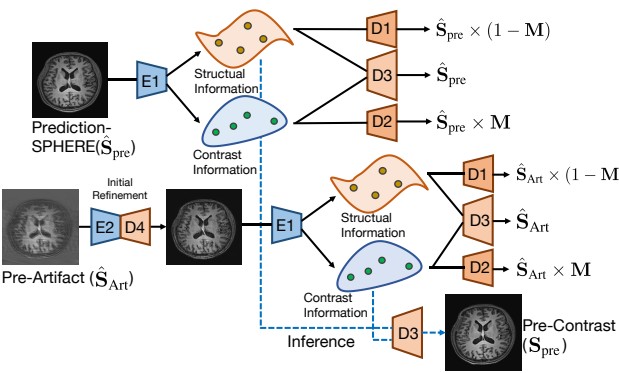

*Figure 3.* The workflow of the SPHERE-Art for artifact correction with the corrupted image. It generates seven outputs that map to corresponding images.

11:

$$\mathbf{S}_{\text{pre}} = f_{\text{inpaint+}}(\mathbf{S}_{\text{post}}, \mathbf{M}; \theta_{\text{inpaint+}}), \quad (24)$$

where $\theta_{\text{inpaint+}}$ are the learnable parameters.

The training loss for the inpainting process is a combination of L1, Structural Similarity Index (SSIM), perceptual, and adversarial losses. The L1 loss minimizes the absolute pixel-wise difference; SSIM Loss penalizes deviations in image structure; Perceptual loss compares the high-level features extracted from a pretrained network (e.g., VGG); The adversarial loss refine the image realism. The detailed equations for all losses are listed in Appendix. B.

### 2.3. Adaption on Artifact Removal

The proposed SPHERE model targets the synthesis of Pre-Contrast images from Post-Contrast images, specifically addressing scenarios where data is missing or corrupted due to heavy motion, aliasing, or zipper artifacts. In some instances, even structurally degraded Pre-Contrast images may still contain valuable information that can be leveraged to enhance the synthesis process. This scenario can also be viewed as an artifact removal task (Cui et al., 2023; Zaitsev et al., 2015), where the desired Pre-Contrast images exist in a compromised form.

To handle such cases, we introduce an artifact removal variant of SPHERE, refered to as SPHERE-Art, designed to effectively extract and integrate useful information from both synthesized and corrupted images. Since the synthesized Pre-Contrast images provide accurate structural representations, we focus on leveraging contrast information from the corrupted images. The architecture of this enhanced model is shown in Fig. 3.

For the synthesized Pre-Contrast image $\hat{\mathbf{S}}_{\text{pre}}$, we employ an encoder $\mathbf{E}_1$ to decompose its manifold representation into

structural and contrast information. Three decoders ($\mathbf{D}_1$, $\mathbf{D}_2$, $\mathbf{D}_3$) are then used to reconstruct the structural domain, contrast domain, and complete image domain, respectively. For the corrupted Pre-Contrast image $\mathbf{S}_{\text{Art}}$, an initial refinement is performed using an encoder-decoder architecture. Subsequently, $\mathbf{S}_{\text{Art}}$ undergoes the same decomposition and reconstruction process as $\hat{\mathbf{S}}_{\text{pre}}$. Finally, a dedicated skip connection is incorporated to merge the structural information from $\hat{\mathbf{S}}_{\text{pre}}$ with the refined contrast information from $\mathbf{S}_{\text{Art}}$, generating the final prediction of the Pre-Contrast image.

A major challenge lies in effectively disentangling structural and contrast information. To ensure robust decomposition and fusion, our model incorporates the seven routes with $L_1$ and SSIM losses. These outputs are meticulously designed to guide the separation of structural and contrast components while promoting their synergistic integration.

## 3. Experiments and Results

**Dataset:** With IRB approval and informed consent, we retrospectively collected 126 cases from Site A (Gadoterate meglumine, 113 training, 13 testing) and 159 cases from Site B (Gadobenate dimeglumine, 149 training, 10 testing). Tab. 1 provides cohort details. Clinical indications included suspected tumors, post-operative follow-ups, and routine brain imaging. Each patient had paired 3D T1-weighted MPRAGE scans for Pre-Contrast, Low-Dose, and Post-Contrast imaging. Images were mean-normalized and affine-registered, using SimpleElastix (Marstal et al., 2016), with the Pre-Contrast image as the reference.

*Table 1.* Dataset cohort description

| Site | Total Cases | Gender | Age | Scanner | Field Strength | TE (sec) | TR (sec) | Flip Angle |
|------|-------------|--------|-----|---------|----------------|----------|----------|------------|
| Site A | 126 | 55 Females 71 Males | $48 \pm 16$ | Philips Insignia | 3T | 2.97-3.11 | 6.41-6.70 | 8° |
| Site B | 159 | 78 Females 81 Males | $52 \pm 17$ | GE Discovery | 3T | 2.99-5.17 | 7.73-12.25 | 8-20° |

**Implementation details:** All experiments were conducted with four NVIDIA A100 40GB GPU on a Intel(R) Xeon(R) CPU E5-2698 v4. The base models of SPHERE used for segmentation, autoencoder, and inpainting+ are all NestedUNet (Zhou et al., 2018) models. To benchmark SPHERE against state-of-the-art methods, we compare its performance with UNet (Ronneberger et al., 2015), ATT-UNet (Oktay et al., 2018), NestedUNet (Zhou et al., 2018), SwinIR (Liang et al., 2021), UKAN (Li et al., 2024), MambaIR (Guo et al., 2025), and BICEPS (Xue et al., 2022). These methods represent the golden standard approaches in image enhancement, segmentation, and synthesis. Each model was trained using its official codes, with hyperparameters such as learning rate, batch size, and loss weights fine-tuned to achieve optimal performance.

Several metrics are employed to quantitatively evaluate im-

age quality, including *Peak Signal-to-Noise Ratio* (PSNR), *Structural Similarity Index Measure* (SSIM), *Contrast-to-Noise Ratio* (CNR), and *Learned Perceptual Image Patch Similarity* (LPIPS) (Zhang et al., 2018). PSNR quantifies image fidelity by comparing signal strength to noise, while SSIM assesses structural similarity between images. CNR evaluates contrast relative to background noise, particularly in regions of interest, and LPIPS leverages deep learning to measure perceptual similarity in a feature space.

### 3.1. Comparative Results

Qualitative results in Fig. 4 show that all existing models contribute to Pre-Contrast image synthesis, producing images that visually resemble the target. However, these methods struggle to balance structural and contrast information, leading to inaccuracies in fine details. In contrast, the proposed SPHERE model demonstrates superior performance across both datasets. As shown in Fig. 4(A1, A2), SPHERE accurately reconstructs contrast-enhanced regions, even in small structures like the choroid plexus, where other methods fail. Additionally, SPHERE preserves structural details highly consistent with the input images, sometimes even surpassing standard-of-care (SOC) Pre-Contrast images due to alignment or tissue dynamics between acquisitions. Similarly, Fig. 4(B1, B2) highlights SPHERE's ability to synthesize tumor structures with high fidelity, outperforming competing methods.

Further validation is provided in Appendix Fig. 9, where SPHERE achieves the closest line profile relative to the true Pre-Contrast image. Quantitative results in Tab. 2 confirm SPHERE's superiority, achieving the highest scores across four image quality metrics on two datasets. Although advanced models like SwinIR, UKAN, and MambaIR contribute to this task, they fall short due to the limitations of direct learning (discussed in Section C). In summary, SPHERE is the only model capable of generating high-quality Pre-Contrast images with precise contrast and structural details, from small structures like the choroid plexus to large pathological regions such as tumors.

**Contrast information Evaluations**

To further assess the accuracy and consistency of synthesized contrast in medical images, we implemented a set of metrics designed to evaluate contrast preservation, fidelity, and structural integrity within the region of interest (ROI), as defined in Eq. 22. Specifically, four metrics were employed for contrast region evaluation: *Contrast Fidelity Score* (CFS), *Gradient Magnitude Similarity Deviation* (GMSD), *Intensity Range Consistency* (IRC), and *Edge Intensity Similarity* (EIS). The mathematical formulations of these metrics are detailed in Appendix A.

As shown in Fig. 5(A), the proposed SPHERE model

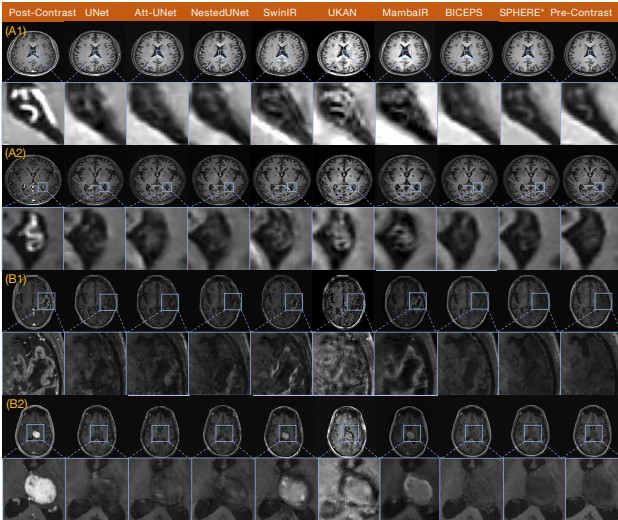

*Figure 4.* The comparative results of representative slices for different methods. (A1,A2) are two representative slices from Site A; (B1,B2) are another two slices from Site B.

achieves the closest intensity match to the SOC Pre-Contrast image, whereas other methods exhibit significant structural inconsistencies. Additionally, the Noise Power Spectrum (NPS) analysis confirms that SPHERE produces the lowest noise levels among all compared methods. Quantitative evaluation in Tab. 2 further validates the superior performance of SPHERE across all contrast metrics, demonstrating its effectiveness in reconstructing contrast region while maintaining high structural accuracy.

### 3.2. Motion Artifact Removal

Models from the UNet family are widely used as backbone architectures for MRI artifact removal, addressing motion artifacts, aliasing, and ringing artifacts (Yang et al., 2017; Chen et al., 2023; Kang & Lee, 2024). To comprehensively evaluate the SPHERE-Art framework, we compared it with state-of-the-art UNet-based models, including UNet (Ronneberger et al., 2015), R2-AttUNet (Alom et al., 2018), and NestedUNet (Zhou et al., 2018). For these baselines, both Post-Contrast images and corrupted Pre-Contrast images were used as inputs to predict clean Pre-Contrast images.

As shown in Fig. 6, all models demonstrate some ability to reduce artifacts, but SPHERE-Art consistently outperforms them, achieving superior artifact correction across all tested scenarios. While residual artifacts remain in other methods, SPHERE-Art reconstructs cleaner images with minimal distortions. Tab. 3 further validates SPHERE-Art's effectiveness, achieving the highest performance scores across all metrics. These results highlight the advantage of domain-specific knowledge integration and the dual-stage learning

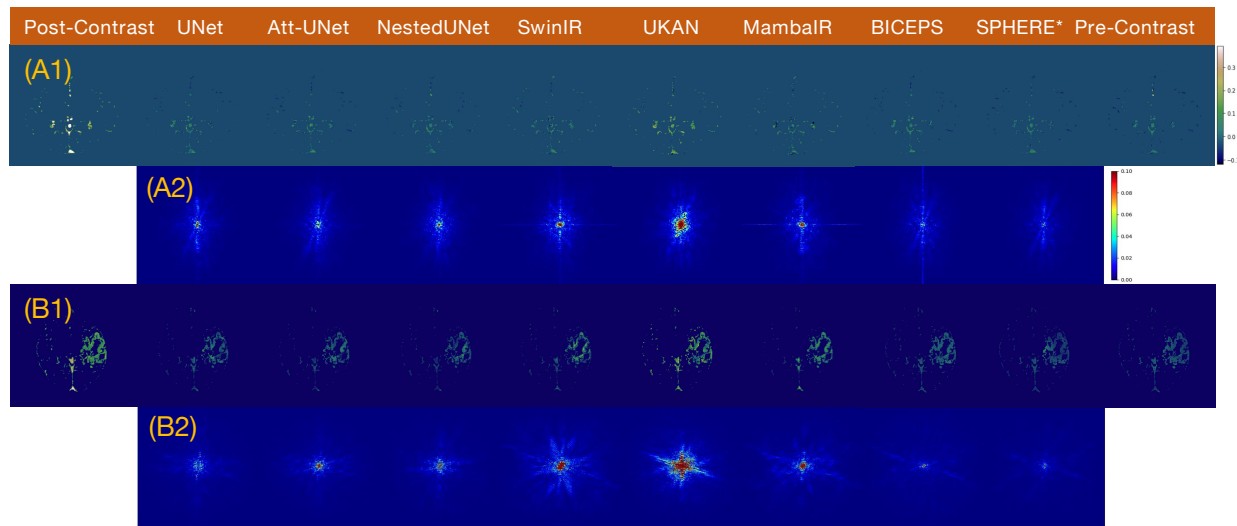

*Figure 5.* The contrast uptake of the representative results from Site A and B; The corresponding NPS maps for each method.

*Table 2.* Quantitative evaluation results of different base methods on the test cases. Bold-faced numbers indicate the best results.

| Method | Site A | | | | Site B | | | |
|---|---|---|---|---|---|---|---|---|
| | PSNR (dB)↑ | SSIM↑ | CNR↓ | LPIPS↓ | PSNR (dB)↑ | SSIM↑ | CNR↓ | LPIPS↓ |
| UNet | $34.6430 \pm 0.22$ | $0.8633 \pm 0.004$ | $0.0819 \pm 0.002$ | $0.0497 \pm 0.002$ | $32.8615 \pm 0.24$ | $0.8943 \pm 0.004$ | $0.0920 \pm 0.002$ | $0.0621 \pm 0.002$ |
| Att-UNet | $34.8340 \pm 0.18$ | $0.8658 \pm 0.003$ | $0.0804 \pm 0.001$ | $0.0507 \pm 0.001$ | $32.8775 \pm 0.16$ | $0.8969 \pm 0.003$ | $0.0906 \pm 0.001$ | $0.0628 \pm 0.002$ |
| UNet++ | $34.5522 \pm 0.10$ | $0.8580 \pm 0.003$ | $0.0836 \pm 0.001$ | $0.0479 \pm 0.002$ | $32.9857 \pm 0.14$ | $0.9002 \pm 0.003$ | $0.0909 \pm 0.001$ | $0.0505 \pm 0.001$ |
| SwinIR | $32.7531 \pm 0.22$ | $0.8124 \pm 0.004$ | $0.1063 \pm 0.002$ | $0.0660 \pm 0.002$ | $31.3174 \pm 0.21$ | $0.8618 \pm 0.004$ | $0.1143 \pm 0.002$ | $0.0679 \pm 0.002$ |
| UKAN | $29.0574 \pm 0.31$ | $0.7833 \pm 0.005$ | $0.0990 \pm 0.002$ | $0.0736 \pm 0.002$ | $24.1449 \pm 0.34$ | $0.6716 \pm 0.005$ | $0.1408 \pm 0.002$ | $0.1148 \pm 0.002$ |
| MambaIR | $32.0594 \pm 0.29$ | $0.8183 \pm 0.004$ | $0.1073 \pm 0.002$ | $0.0867 \pm 0.002$ | $29.7819 \pm 0.29$ | $0.8151 \pm 0.004$ | $0.1375 \pm 0.002$ | $0.1121 \pm 0.002$ |
| BICEPS | $35.2499 \pm 0.23$ | $0.8735 \pm 0.004$ | $0.0751 \pm 0.001$ | $0.0397 \pm 0.001$ | $34.5037 \pm 0.19$ | $0.9122 \pm 0.003$ | $0.0772 \pm 0.001$ | $0.0381 \pm 0.001$ |
| SPHERE* | $\mathbf{36.8244} \pm 0.14$ | $\mathbf{0.9026} \pm 0.002$ | $\mathbf{0.0628} \pm 0.001$ | $\mathbf{0.0313} \pm 0.001$ | $\mathbf{35.3617} \pm 0.15$ | $\mathbf{0.9263} \pm 0.001$ | $\mathbf{0.0699} \pm 0.002$ | $\mathbf{0.0330} \pm 0.001$ |
| Method | GMSD ↓ | CFS↑ | IRC↑ | EIS↑ | GMSD ↓ | CFS↑ | IRC↑ | EIS↑ |
| UNet | $0.0360 \pm 0.002$ | $0.8581 \pm 0.005$ | $0.8069 \pm 0.009$ | $0.3886 \pm 0.007$ | $0.0265 \pm 0.001$ | $0.7394 \pm 0.005$ | $0.5747 \pm 0.008$ | $0.2183 \pm 0.009$ |
| Att-UNet | $0.0367 \pm 0.003$ | $0.8641 \pm 0.005$ | $0.7976 \pm 0.008$ | $0.3765 \pm 0.012$ | $0.0256 \pm 0.002$ | $0.8633 \pm 0.004$ | $0.7013 \pm 0.009$ | $0.2151 \pm 0.014$ |
| UNet++ | $0.0362 \pm 0.003$ | $0.8462 \pm 0.005$ | $0.8454 \pm 0.007$ | $0.3746 \pm 0.007$ | $0.0258 \pm 0.002$ | $0.8441 \pm 0.005$ | $0.6547 \pm 0.007$ | $0.2389 \pm 0.010$ |
| SwinIR | $0.0402 \pm 0.003$ | $0.6081 \pm 0.005$ | $0.6675 \pm 0.010$ | $0.2821 \pm 0.015$ | $0.0326 \pm 0.003$ | $0.7421 \pm 0.005$ | $0.3350 \pm 0.012$ | $0.0290 \pm 0.013$ |
| UKAN | $0.0601 \pm 0.003$ | $0.2937 \pm 0.005$ | $0.0002 \pm 0.010$ | $0.2339 \pm 0.016$ | $0.0777 \pm 0.004$ | $-0.7269 \pm 0.006$ | $-1.2471 \pm 0.012$ | $0.0357 \pm 0.015$ |
| MambaIR | $0.0441 \pm 0.002$ | $0.4989 \pm 0.004$ | $0.5902 \pm 0.009$ | $0.1886 \pm 0.015$ | $0.0403 \pm 0.002$ | $0.8164 \pm 0.003$ | $-0.0469 \pm 0.009$ | $-0.0696 \pm 0.014$ |
| BICEPS | $0.0349 \pm 0.002$ | $0.8857 \pm 0.002$ | $0.8535 \pm 0.005$ | $0.4253 \pm 0.006$ | $0.0241 \pm 0.002$ | $0.8595 \pm 0.002$ | $0.8040 \pm 0.005$ | $0.3619 \pm 0.007$ |
| SPHERE* | $\mathbf{0.0315} \pm 0.001$ | $\mathbf{0.9181} \pm 0.002$ | $\mathbf{0.8684} \pm 0.006$ | $\mathbf{0.5364} \pm 0.005$ | $\mathbf{0.0221} \pm 0.001$ | $\mathbf{0.8803} \pm 0.002$ | $\mathbf{0.8410} \pm 0.005$ | $\mathbf{0.4433} \pm 0.005$ |

*Table 3.* Quantitative evaluation results of different base methods artifacts removal. Bold-faced numbers indicate the best results.

| Method | PSNR (dB)↑ | SSIM↑ | RMSE↓ | CNR↓ | LIPIPS ↓ |
|---|---|---|---|---|---|
| Pre-Artifact | 22.7495 | 0.1089 | 0.1167 | 0.3617 | 0.0298 |
| UNet | 33.4898 | 0.8839 | 0.0213 | 0.0566 | 0.0390 |
| R2-AttUNet | 33.5804 | 0.8908 | 0.0210 | 0.0557 | 0.0308 |
| NestedUNet | 33.0154 | 0.8761 | 0.0225 | 0.0592 | 0.0304 |
| SPHERE-Art* | **37.1245** | **0.9063** | 0.0226 | 0.0612 | 0.0298 |

framework, positioning SPHERE-Art as a leading approach for artifact removal in clinical MRI applications.

## 4. Downstreaming Tasks

To further evaluate the clinical significance and quality of the synthesized Pre-Contrast images, we assess their impact on key downstream tasks, including dose simulation, contrast enhancement, and applications in spine and breast imaging. All these tasks are essential for contrast-enhanced MRI applications.

**Low Dose Simulation:** Low-dose simulation is a critical task for data augmentation and optimizing radiation exposure in clinical settings, adhering to the 'As Low As Reasonably Achievable' (ALARA) principle (Sourbron & Buckley, 2011; Wang et al., 2023). This process enables the determination of the minimal effective contrast agent dosage, improving patient safety while preserving diagnostic quality. Low-dose simulations are performed using both Pre-Contrast and Post-Contrast images. The quantitative evaluations reveal that using synthesized Pre-Contrast images deliver results comparable to using SOC Pre-Contrast

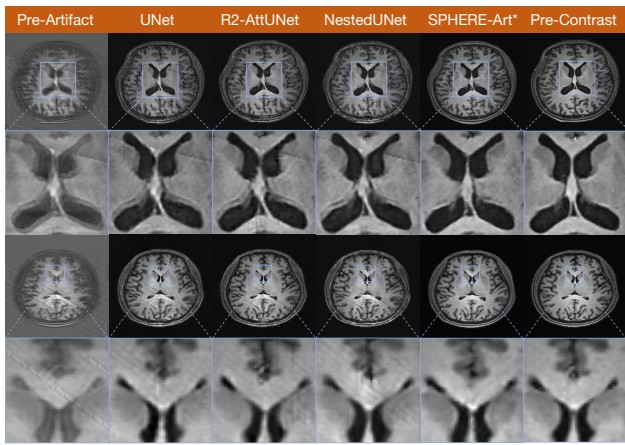

*Figure 6.* The comparative results on the artifact removal of different methods.

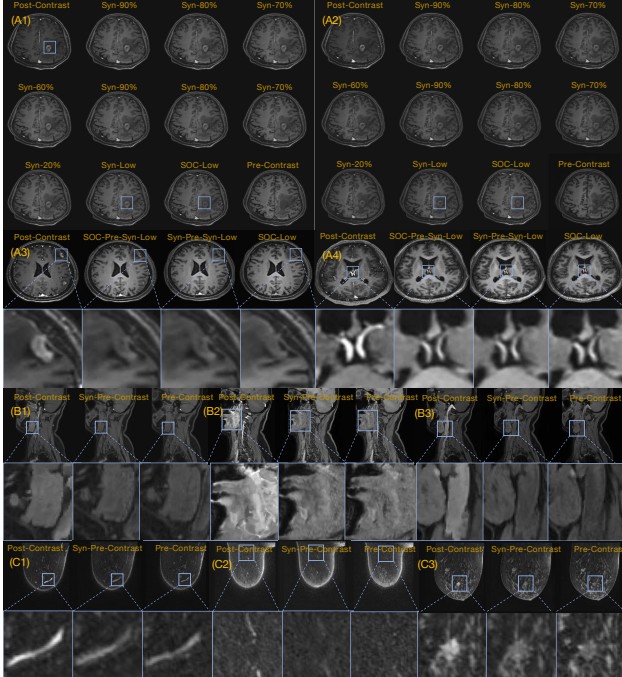

*Figure 7.* The results of the dowmstreaming tasks. (A1) is the low-dose simulation results with sythesized Pre-Contrast; (A2) is the simulation results with SOC Pre-Contrast; (A3,A4) are two comparative results on 10% dose; (B1-B3) are three slices on Spine images; (C1-C3) are slices on Breast images. (It's better to view in a zoomed fashion).

images. As shown in Fig. 7(A1-A4), the overall visual quality of the synthesized low-dose images closely aligns with that of the SOC-based counterparts, demonstrating the feasibility of the proposed approach in generating reliable Low-dose images for clinical applications.

*Table 4.* Quantitative evaluation results of different base methods on the test cases. Bold-faced numbers indicate the best results.

| Tasks | Method | PSNR (dB)↑ | SSIM↑ | RMSE↓ | CNR↓ | LIPIPS ↓ |
|---|---|---|---|---|---|---|
| Dose Simulation | SOC-Pre-Contrast | 39.6535 | 0.9630 | 0.1736 | 0.0036 | 0.0215 |
| | Syn-Pre-Contrast* | 37.1081 | 0.9454 | 0.2418 | 0.0381 | 0.0388 |
| Spine Application | Brain transferred* | 36.5489 | 0.9042 | 0.0238 | 0.0686 | 0.0496 |
| Breast Application | Brain transferred* | 39.4793 | 0.8472 | 0.0525 | 0.1620 | 0.1646 |

**Spine Application**: In spine imaging, contrast-enhanced MRI assesses conditions like infections, tumors, and degenerative diseases, with Pre-Contrast images crucial for distinguishing normal tissue from pathological enhancement and subtle marrow changes (Breger et al., 1989; Colosimo et al., 2006). A Post-to-Pre reconstruction technique reduces scan time, benefiting patients prone to motion artifacts, such as the elderly and those with chronic pain. To assess the adaptability of our method, we fine-tuned a pretrained model (trained on Site A data) using a limited Spine imaging dataset consisting of T1-weighted Pre-Contrast and Post-Contrast scans from five patients. As shown in Tab. 4 (quantitative) and Fig. 7(B1-B3) (qualitative), the transfer-learned model effectively synthesizes high-quality Pre-Contrast images, demonstrating strong generalization to Spine imaging.

**Breast Application**: Pre-Contrast images are essential for baseline signal evaluation in contrast-enhanced Breast MRI (Mann et al., 2019; Van Nijnatten et al., 2024). However, these images often suffer from artifacts and misalignment issues that cannot be resolved through registration. A Post-to-Pre reconstruction approach addresses this limitation. Using

a pretrained model from Site A, we fine-tuned it on breast imaging data from just three cases over 10,000 training iterations. Qualitative results in Fig. 7 (C1-C3) show that the synthesized images closely resemble the SOC images, while quantitative results in Tab. 4 further highlight strong performance metrics, underscoring the approach's capability to enhance diagnostic quality and workflow efficiency.

## 5. Ablation Study

In this section, we present an ablation study to assess the contribution of key components in the proposed method. Specifically, we investigate the effects of the dual-stage learning framework, the hyperintensity prior, the arbitrary segmentation branch, the autoencoder module, and the selection of the brightness threshold. Both the quantitative results (Tab. 5) and the qualitative findings (Fig. 8) demonstrate that each of these components plays an essential role in improving model performance. In particular, the dual-stage learning strategy, hyperintensity prior, and arbitrary segmentation branch contribute significantly. Removing any of these components results in a consistent performance decline across multiple evaluation metrics, underscoring their

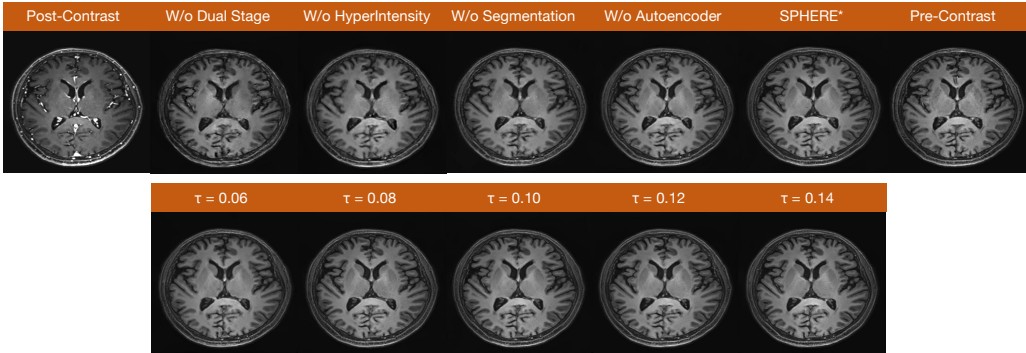

*Figure 8.* The results of the ablation study on the dual-stage learning framework, hyperintensity prior, arbitrary segmentation branch, autoencoder module, and the choice of brightness threshold $\tau$.

*Table 5.* Ablation study on different components and $\tau$ selection results

| Configuration | PSNR ↑ | SSIM ↑ | CNR ↓ | LPIPS ↓ | GMSD ↓ | CFS ↑ | IRC ↑ | EIS ↑ |
|---|---|---|---|---|---|---|---|---|
| W/o Dual Stage | 34.5522 | 0.8580 | 0.0836 | 0.0479 | 0.0362 | 0.8462 | 0.8454 | 0.3746 |
| W/o Hyperintensity | 35.7608 | 0.8842 | 0.0729 | 0.0434 | 0.0356 | 0.8970 | 0.8252 | 0.4041 |
| W/o Segmentation | 36.3764 | 0.8951 | 0.0666 | 0.0346 | 0.0337 | 0.9068 | 0.8558 | 0.4703 |
| W/o Autoencoder | 36.6764 | 0.9006 | 0.0635 | 0.0321 | 0.0328 | 0.9119 | 0.8585 | 0.5075 |
| SPHERE* | **36.8244** | **0.9026** | **0.0628** | **0.0313** | **0.0315** | **0.9181** | **0.8684** | **0.5364** |
| Brightness threshold $\tau$ selection | | | | | | | | |
| $\tau = 0.06$ | 35.7835 | 0.8990 | 0.0651 | 0.0366 | 0.0320 | 0.9109 | 0.8560 | **0.5366** |
| $\tau = 0.08$ | **36.8841** | 0.9034 | 0.0628 | **0.0298** | **0.0315** | 0.9179 | 0.8676 | 0.5319 |
| $\tau = 0.10$ | 36.8244 | 0.9026 | 0.0628 | 0.0313 | **0.0315** | **0.9181** | 0.8684 | 0.5364 |
| $\tau = 0.12$ | 36.8759 | **0.9035** | **0.0624** | 0.0299 | 0.0317 | 0.9167 | 0.8687 | 0.5333 |
| $\tau = 0.14$ | 36.7256 | 0.9008 | 0.0635 | 0.0313 | 0.0320 | 0.9097 | **0.8702** | 0.5246 |

importance within the overall framework.

For the brightness threshold $\tau$, we evaluated a range of values from 0.06 to 0.14. The results indicate that values within the studied range consistently yield strong performance with only minor variation. For instance, all tested values in this range achieve a PSNR of approximately 36.80±0.08 and an SSIM of approximately 0.90±0.003. These findings suggest that the model is relatively robust to the specific choice of brightness threshold, thereby simplifying hyperparameter tuning and improving the method's general applicability.

## 6. Conclusion

The proposed method offers a novel solution for synthesizing Pre-Contrast images directly from Post-Contrast data, addressing key challenges in medical imaging where Pre-Contrast images are often unavailable, misaligned, or corrupted. This significant advancement enables high-accuracy Pre-Contrast synthesis, particularly excelling in reconstructing large tumor regions, thus enhancing diagnostic reliability. By potentially eliminating the need for Pre-Contrast acquisition during MRI, this method reduces examination time, improving clinical efficiency in high-demand settings. Furthermore, it minimizes patient discomfort and exposure

to prolonged scanning, contributing to a more streamlined and patient-centered imaging process.

## Impact Statement

This paper presents the first MRI physics-guided framework for clinically viable Pre-Contrast image synthesis, addressing a key challenge in contrast-enhanced MRI. By generating high-fidelity Pre-Contrast images, it overcomes barriers posed by missing or corrupted scans, improves workflow efficiency, reduces costs, and offers a transformative solution for clinical practice.

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

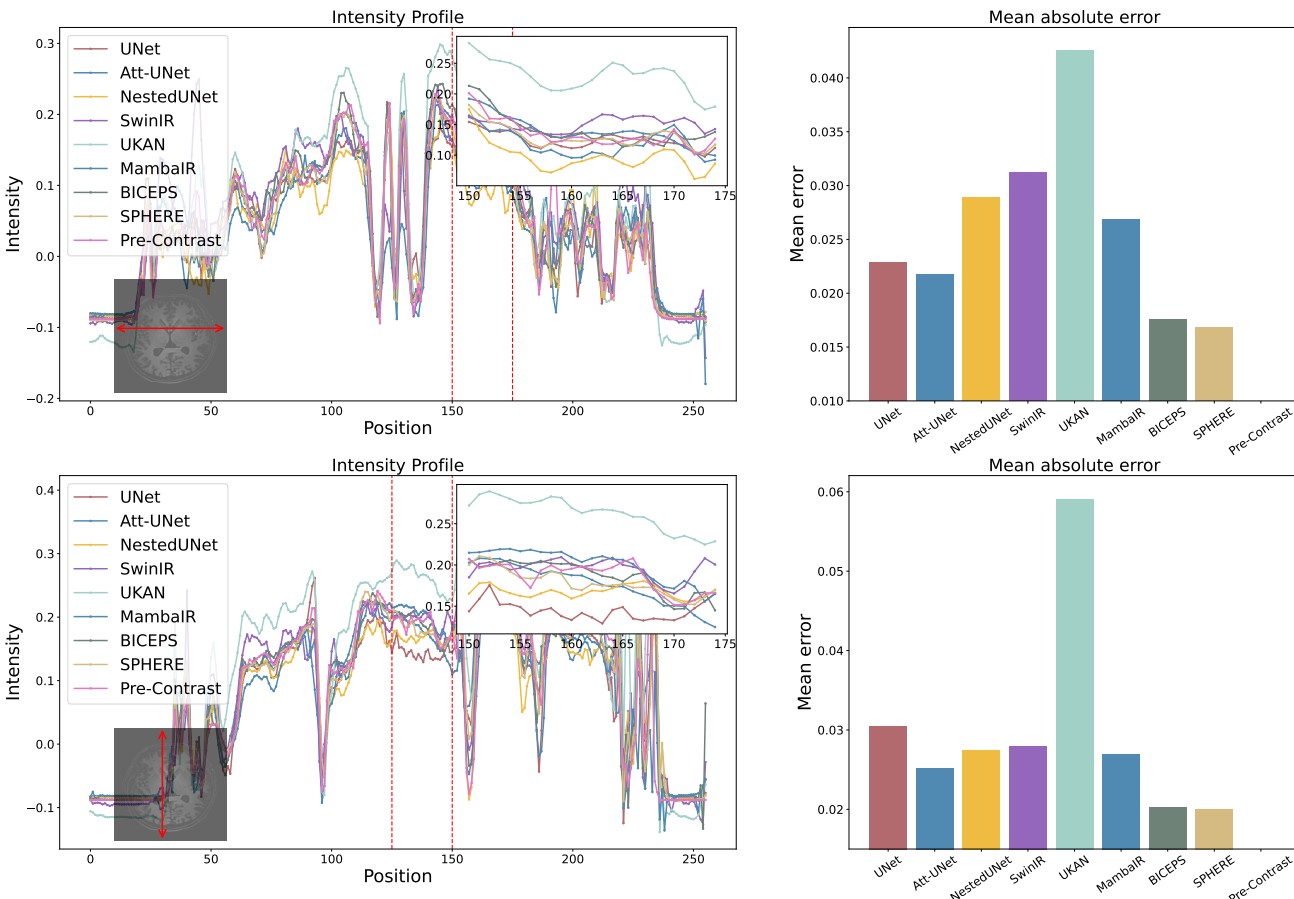

*Figure 9.* The intensity profile of a representative slice for different methods. The left two plots depict the line profile of a representative slice on both vertical and horizontal orientations. the left two figures illustrate the mean absolute errors of different methods relative to the SOC Pre-Contrast image.

# Appendix

## A. Contrast Evaluation Metric

Four quantitative metrics for contrast area evaluation are utilized:

1). *Contrast Fidelity Score* (CFS) measures the preservation of contrast magnitude between the ground truth and prediction. It is defined as:

$$\text{CFS} = 1 - \frac{\left| \mu_{\hat{\mathbf{S}}_{\text{pre}}} - \mu_{\mathbf{S}_{\text{pre}}} \right|}{\mu_{\mathbf{S}_{\text{pre}}} + \epsilon}, \tag{25}$$

where $\mu_{\hat{\mathbf{S}}_{\text{pre}}}$ and $\mu_{\mathbf{S}_{\text{pre}}}$ are the mean intensity value of prediction and ground truth images. $\epsilon$ is a minimal value to avoid zero denominator. This metric measures differences in the mean intensities within the masked region, with higher values indicating better contrast preservation.

2). *Gradient Magnitude Similarity Deviation* (GMSD) measures the similarity between the gradient magnitudes of the prediction and ground truth images within the masked region. It is defined as:

$$\text{GMSD} = \sqrt{\frac{1}{N} \sum_{i=1}^{N} \left( g_{\mathbf{S}_{\text{pre}}}^{i} - g_{\hat{\mathbf{S}}_{\text{pre}}}^{i} \right)^2}, \tag{26}$$

where $N$ is the number of samples. $g_{\mathbf{S}_{\text{pre}}}$ and $g_{\hat{\mathbf{S}}_{\text{pre}}}$ are the gradient magnitudes of the ground truth and prediction images, respectively, computed using the Sobel filter (Kanopoulos et al., 1988).

3). *Intensity Range Consistency* (IRC) evaluates the agreement in intensity range ($\Delta I$) between the prediction and ground truth. It is computed as:

$$\text{IRC} = 1 - \frac{\left| \Delta I_{\hat{\mathbf{S}}_{\text{pre}}} - \Delta I_{\mathbf{S}_{\text{pre}}} \right|}{\Delta I_{\mathbf{S}_{\text{pre}}}}, \tag{27}$$

where $\Delta I = \max(I) - \min(I)$ is the peak-to-peak intensity range.

4). *Edge Intensity Similarity* (EIS) assesses the correlation between edge intensities of the ground truth and prediction within the masked region. It is calculated using the Pearson correlation coefficient (Freedman et al., 2007):

$$\text{EIS} = \text{corr}(E_{\mathbf{S}_{\text{pre}}}, E_{\hat{\mathbf{S}}_{\text{pre}}}) \tag{28}$$

where $E_{\mathbf{S}_{\text{pre}}}$ and $E_{\hat{\mathbf{S}}_{\text{pre}}}$ are the edge intensity maps of the ground truth and prediction, respectively, obtained using the Sobel filter(Kanopoulos et al., 1988). EIS ranges from $-1$ to $1$, with values closer to $1$ indicating better similarity.

## B. Training Loss for the SPHERE

The training loss for the inpainting+ process is a combination of L1, Structural Similarity Index (SSIM), perceptual, and adversarial losses. The $L_1$ loss minimizes the absolute difference between the predicted pre-contrast image $\hat{\mathbf{S}}_{\text{pre}}$ and the ground truth $\mathbf{S}_{\text{pre}}$; SSIM Loss measures the structural similarity between the predicted and ground truth images which penalizes deviations in image structure; Perceptual loss compares the high-level features of the predicted and ground truth images, extracted from a pretrained network (e.g., VGG (Simonyan & Zisserman, 2015)); The adversarial loss comes from a GAN (Goodfellow et al., 2014), where a generator predicts $\hat{\mathbf{S}}_{\text{pre}}$ and a discriminator $\mathbf{D}$ distinguishes between real $\mathbf{S}_{\text{pre}}$ and fake $\hat{\mathbf{S}}_{\text{pre}}$:

$$\begin{cases} \mathcal{L}_{\text{L1}} = \dfrac{1}{N} \sum \left| \mathbf{S}_{\text{pre}} - \hat{\mathbf{S}}_{\text{pre}} \right|, \\[2mm] \mathcal{L}_{\text{SSIM}} = 1 - \dfrac{(2\mu_{\mathbf{S}_{\text{pre}}}\mu_{\hat{\mathbf{S}}} + C_1)(2\sigma_{\mathbf{S}_{\text{pre}},\hat{\mathbf{S}}} + C_2)}{(\mu_{\mathbf{S}_{\text{pre}}}^2 + \mu_{\hat{\mathbf{S}}}^2 + C_1)(\sigma_{\mathbf{S}_{\text{pre}}}^2 + \sigma_{\hat{\mathbf{S}}}^2 + C_2)}, \\[2mm] \mathcal{L}_{\text{perceptual}} = \sum_l \dfrac{1}{N_l} \| \phi_l(\mathbf{S}_{\text{pre}}) - \phi_l(\hat{\mathbf{S}}_{\text{pre}}) \|^2, \\[2mm] \mathcal{L}_{\text{adv}} = -\dfrac{1}{N} \sum \log(\mathbf{D}(\hat{\mathbf{S}}_{\text{pre}})), \end{cases} \tag{29}$$

where $C_1$ and $C_2$ are small value to avoid zero denominator; $\sigma_{\mathbf{S}_{\text{pre}}}^2$ and $\sigma_{\hat{\mathbf{S}}}^2$ are the variance of $\mathbf{S}_{\text{pre}}$ and $\hat{\mathbf{S}}$; $\sigma_{\mathbf{S}_{\text{pre}},\hat{\mathbf{S}}}$ is the covariance between the $\mathbf{S}_{\text{pre}}$ and $\hat{\mathbf{S}}$; $N_l$ is the number of feature map layers for perceptual loss calculation; $\phi_l$ is the network used for feature extration (VGG).

For the discriminator $\mathbf{D}$:

$$\mathcal{L}_{\text{adv}}^{\text{disc}} = -\frac{1}{N} \sum \left[ \log(\mathbf{D}(\mathbf{S}_{\text{pre}})) + \log \left( 1 - \mathbf{D}(\hat{\mathbf{S}}_{\text{pre}}) \right) \right]. \tag{30}$$

In summary, the total inpainting+ loss combines all these components in the two stages with appropriate weighting factors:

$$\mathcal{L}_{\text{Inpaint+}} = \lambda_{\text{L1}} \mathcal{L}_{\text{L1}} + \lambda_{\text{SSIM}} \mathcal{L}_{\text{SSIM}} + \lambda_{\text{perceptual}} \mathcal{L}_{\text{perceptual}} + \lambda_{\text{adv}} \mathcal{L}_{\text{adv}}^{\text{gen}}, \tag{31}$$

Where $\lambda_{\text{L1}} = 1$, $\lambda_{\text{SSIM}} = 10$, $\lambda_{\text{perceptual}} = 0.5$, and $\lambda_{\text{adv}} = 1$ are weights to balance the contributions of different loss terms. The discriminator loss $\mathcal{L}_{\text{adv}}^{\text{disc}}$ is used to update the discriminator separately during adversarial training.

The segmentation stage is optimized using two loss functions: L1 loss for autoencoder reconstruction and BCE loss for segmentation:

$$\begin{cases} \mathcal{L}_{\text{ae}} = \dfrac{1}{N} \sum \left| \mathbf{S}_{\text{post}} - \hat{\mathbf{S}}_{\text{post}} \right|, \\ \mathcal{L}_{\text{bce}} = -\dfrac{1}{N} \sum \left[ \mathbf{M}_{\text{true}} \log(\mathbf{M}) + (1 - \mathbf{M}_{\text{true}}) \log\left(1 - \mathbf{M}\right) \right]. \end{cases} \tag{32}$$

Moreover, two penalty terms are incorporated based on the MRI physics of the synthesis defined in Eq. 23 and 11:

$$\begin{cases} \mathcal{R}_{\text{hyper}} = \dfrac{1}{N} \sum \left| \mathbf{M} - \sigma(u(\mathbf{S}'_{\text{post}} - \beta_0)) \right|^2 \\ \mathcal{R}_{\text{psyn}} = \dfrac{1}{N} \sum \left| \hat{\mathbf{S}}_{\text{pre}} - \dfrac{\rho \sin(\alpha) \left( \mathbf{S}_{\text{post}} e^{\text{TR} r_1 \beta \mathbf{M}} - \mathbf{S}_{\text{post}} \cos(\alpha) - \rho e^{\text{TR} r_1 \beta \mathbf{M}} \sin(\alpha) + \rho \sin(\alpha) \right)}{\mathbf{S}_{\text{post}} e^{\text{TR} r_1 \beta \mathbf{M}} \cos(\alpha) - \mathbf{S}_{\text{post}} \cos(\alpha) - \frac{\rho e^{\text{TR} r_1 \beta \mathbf{M}} \sin(2\alpha)}{2} + \rho \sin(\alpha)} \right|^2. \end{cases} \tag{33}$$

Here, $\beta_0 = 0.3$, $u = 5$. we define the scale based on the common assumption that proton density $\rho = 1$ in brain ventricles (Mezer et al., 2016).

In summary, the total losses for SPHERE generator are:

$$\mathcal{L}_{\text{total}} = \lambda_{\text{Inpaint+}} \mathcal{L}_{\text{Inpaint+}} + \lambda_{\text{ae}} \mathcal{L}_{\text{ae}} + \lambda_{\text{bce}} \mathcal{L}_{\text{bce}} + \lambda_{\text{hyper}} \mathcal{R}_{\text{hyper}} + \lambda_{\text{psyn}} \mathcal{R}_{\text{psyn}}, \tag{34}$$

where $\lambda_{\text{Inpaint+}} = 1$, $\lambda_{\text{ae}} = 1$, $\lambda_{\text{bce}} = 10$, $\lambda_{\text{hyper}} = 0.1$, and $\lambda_{\text{psyn}} = 0.01$. The weighting parameters were selected based on the inherent characteristics of each loss function, rather than being optimized for specific imaging modalities or anatomical regions. This design choice supports the generalizability of the proposed method across different scanners, clinical sites, and anatomical structures.

## C. Gradient Dynamics and Conflict Resolution in Direct and Two-Stage Learning

### C.1. Challenges in Direct Learning

Direct learning seeks to map $\mathbf{S}_{\text{post}}$ to $\mathbf{S}_{\text{pre}}$ through a single neural network $f_{\text{direct}}$. While the pixel-wise loss function (e.g. *mean squared error* (MSE)) is computed independently, the shared network parameters $\theta_{\text{direct}}$ create dependencies across all pixels during optimization. This parameter sharing leads to challenges in gradient dynamics, particularly in regions with high signal disparity. The optimization goal for direct learning is to minimize the reconstruction loss (e.g. MSE):

$$\mathcal{L}_{\text{direct}} = \dfrac{1}{N} \sum \left| f_{\text{direct}}(\mathbf{S}_{\text{post}}; \theta_{\text{direct}}) - \mathbf{S}_{\text{pre}} \right|^2. \tag{35}$$

The gradient of this loss with respect to the network parameters $\theta_{\text{direct}}$ is given by:

$$\Delta \theta_{\text{direct}} = \frac{\partial \mathcal{L}_{\text{direct}}}{\partial \theta_{\text{direct}}} = \frac{2}{N} \sum (z - \mathbf{S}_{\text{pre}}) \cdot \frac{\partial z}{\partial \theta_{\text{direct}}}, \tag{36}$$

where $z$ is the model output. The term $\mathbf{S}_{\text{post}}$ depends nonlinearly on the contrast concentration $[\text{Gd}]$ due to the relaxation mechanism in T1-weighted imaging:

$$\mathbf{T}_{\text{post}} = \frac{1}{\frac{1}{\mathbf{T}_{\text{pre}}} + r_1 \cdot [\text{Gd}]}, \tag{37}$$

This relationship amplifies the signal difference $\mathbf{S}_{\text{post}} - \mathbf{S}_{\text{pre}}$ in regions with high $[\text{Gd}]$, generating large gradient magnitudes. Conversely, in non-enhanced regions ($[\text{Gd}] = 0$), the gradients are smaller and dominated by intrinsic tissue properties such as $\mathbf{T}_1^{\text{pre}}$ and proton density $\rho$.

The disproportionate gradient contributions from contrast-enhanced regions cause a bias in the parameter updates, which will leads to suboptimal performance, as the network prioritizes fitting high-contrast regions at the expense of accuracy in non-enhanced regions, where smooth tissue-specific variations are critical. This leads to gradient conflicts in the model training process. The gradient for a single pixel is expressed as:

$$\frac{\partial \mathcal{L}_{\text{direct}}}{\partial z} = 2 \cdot (z - \mathbf{S}_{\text{pre}}). \tag{38}$$

In contrast-enhanced regions, the large signal differences caused by the nonlinear dependence on [Gd] dominate the gradient:

$$\mathbf{S}_{\text{post}} - \mathbf{S}_{\text{pre}} \sim \mathcal{O}([\text{Gd}]), \tag{39}$$

whereas in non-enhanced regions, the gradient scales as:

$$\mathbf{S}_{\text{post}} - \mathbf{S}_{\text{pre}} \sim \mathcal{O}(1). \tag{40}$$

This discrepancy is further exacerbated by the shared network parameters, as the gradients from high-contrast regions propagate throughout the network, overshadowing updates for low-contrast areas. The result is a gradient conflict that impedes accurate reconstruction of Pre-Contrast images.

### C.2. Conflict Resolution via Two-Stage Learning

Two-stage learning resolves the gradient conflicts inherent in direct learning by decoupling the tasks of contrast detection and pre-contrast reconstruction. This framework consists of two sequentially optimized components, including a segmentation stage to localize contrast-enhanced regions, and a inpainting+ stage to synthesize the Pre-Contrast image using both the segmentation results and the Post-Contrast image.

### Stage 1: Contrast Segmentation

The segmentation network $f_{\text{seg}}$ learns a binary map $\mathbf{M}$, which identifies regions with significant contrast enhancement. its gradients with BCE loss are constrained to binary classification, avoiding the variance amplification observed in direct learning. The gradient is:

$$\frac{\partial \mathcal{L}_{\text{seg}}}{\partial \mathbf{M}} = \begin{cases} -\frac{1}{\mathbf{M}}, & \mathbf{M} = 1, \\ \frac{1}{1-\mathbf{M}}, & \mathbf{M} = 0. \end{cases} \tag{41}$$

Unlike the gradients in direct learning, which scale with the signal differences $\Delta\mathbf{S} = \mathbf{S}_{\text{post}} - \mathbf{S}_{\text{pre}}$, the segmentation gradients depend solely on classification errors and are bounded. This stability ensures that the network focuses on contrast detection without being dominated by regions with large intensity differences.

### Stage 2: Pre-Contrast Reconstruction

The reconstruction network $f_{\text{inpaint+}}$ synthesizes the Pre-Contrast image $\mathbf{S}_{\text{pre}}$ using both the Post-Contrast image $\mathbf{S}_{\text{post}}$ and the segmentation map $\mathbf{M}$. The reconstruction loss is defined as:

$$\mathcal{L}_{\text{recon}} = \frac{1}{N} \sum \mathbf{M} \cdot \left(f_{\text{inpaint+}}(\mathbf{S}_{\text{post}}, \mathbf{M}; \theta_{\text{recon}}) - \mathbf{S}_{\text{pre}}\right)^2.$$

By incorporating $\mathbf{M}$, the reconstruction task focuses exclusively on contrast-enhanced regions, while gradients for non-enhanced regions ($\mathbf{M} = 0$) are masked. For a pixel in the enhanced region ($\mathbf{M} = 1$), the gradient of the reconstruction loss is:

$$\frac{\partial \mathcal{L}_{\text{recon}}}{\partial \theta_{\text{recon}}} = \frac{2}{N} \sum \mathbf{M} \cdot (z - \mathbf{S}_{\text{pre}}) \cdot \frac{\partial z}{\partial \theta_{\text{recon}}}. \tag{42}$$

This formulation mitigates the gradient conflicts observed in direct learning. Unlike the direct approach, where the gradient magnitude is dominated by $\Delta\mathbf{S}$ in high contrast regions, the two-stage framework ensures that reconstruction gradients are modulated by $\mathbf{M}$, effectively isolating the tasks of contrast detection and Pre-Contrast synthesis.

### C.3. Experimental Results

To further analyze the learning behavior of direct learning methods (e.g., NestedUNet) and the proposed SPHERE framework, we examine their loss curves during training. The L1 and SSIM loss curves in Fig. 10 indicate that direct learning methods tend to converge to suboptimal loss values, whereas SPHERE exhibits a continuous decline in loss, ultimately achieving superior performance.

Additionally, we assess the similarity between the loss curves of contrast and structural regions. To quantify this, we employ *Pearson Correlation* (Freedman et al., 2007), *Cosine Similarity* (Xia et al., 2015), and the *Gradient Similarity Index* (GSI). For contrast loss $l_c$ and structural loss $l_s$, GSI is defined as:

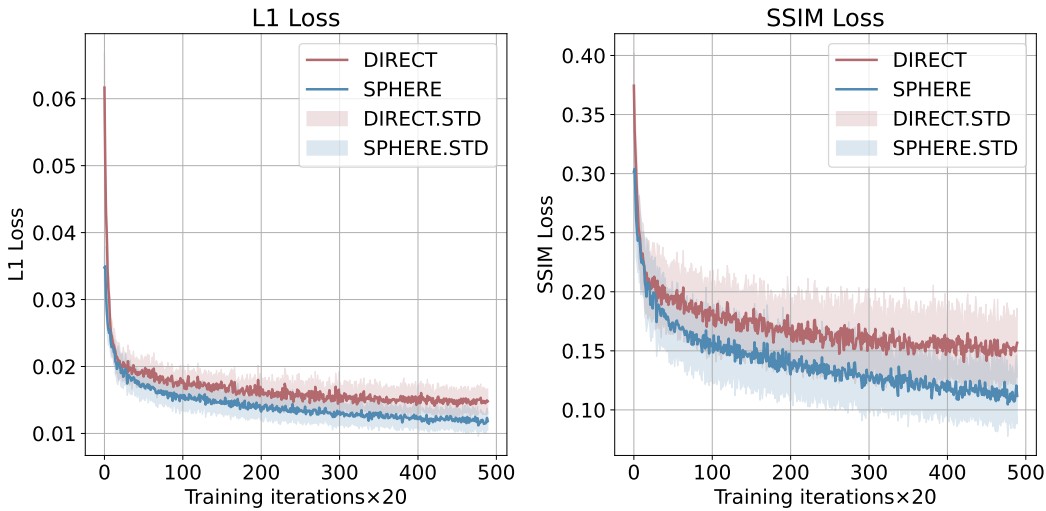

*Figure 10.* The loss comparison between the direct learning method and the proposed SPHERE

$$GSI(l_c, l_s) = \frac{2\sum \nabla l_c \cdot \nabla l_s}{\sum \nabla l_c^2 + \sum \nabla l_s^2}, \tag{43}$$

where $\nabla l_c = \frac{dl_c}{dt}$ and $\nabla l_s = \frac{dl_s}{dt}$ represent their respective gradients. The term $2\sum \nabla l_c \cdot \nabla l_s$ quantifies gradient alignment, while $\sum \nabla l_c^2 + \sum \nabla l_s^2$ normalizes the score.

As shown in Tab. 6, the proposed SPHERE framework achieves significantly higher correlation and similarity in loss optimization between contrast and structural regions, demonstrating a more harmonized learning process. In contrast, direct learning exhibits lower similarity, suggesting potential optimization conflicts between contrast and structural components, as analyzed in Section. C.1.

*Table 6.* Quantitative evaluation of the relationship between contrast and structural region losses, indicating the optimization consistency across different methods.

| Method | Pearson Correlation ↑ | Cosine Similarity↑ | Gradient Similarity Index↑ |
|--------|----------------------|---------------------|-----------------------------|
| DIRECT | 0.5414 | 0.9210 | 0.1549 |
| SPHERE | **0.8226** | **0.9790** | **0.2638** |

## D. Clinical Validation

To comprehensively assess the quality of the synthesized pre-contrast (Syn-Pre) images, we invited two independent radiologists for evaluation. Reader #1 has over 15 years of clinical experience in radiology, and Reader #2 has over 10 years. A total of 15 cases were reviewed, including a range of pathologies such as glioblastoma (GBM), lymphoma (CNS and Hodgkin's-related), anaplastic astrocytoma, meningioma, oligodendroglioma, Von Hippel–Lindau disease, and fungal/parasitic infections. Five evaluation criteria were used: Perceived Image Quality, Anatomical Alignment, Tissue Visualization (usability with Post-Contrast image), Diagnostic Value (when paired with Post-Contrast image), and Imaging Artifacts. Each metric was scored on a 1–4 Likert scale.

Fig. 11 shows that Syn-Pre exhibits a highly similar and authentic pathological appearance compared to SOC-Pre. Quantitatively, Syn-Pre achieved higher scores than SOC-Pre in Perceived Image Quality ($3.96 \pm 0.19$ vs. $3.64 \pm 0.49$), Anatomical Alignment ($3.96 \pm 0.19$ vs. $3.75 \pm 0.52$), and Imaging Artifacts ($4.00 \pm 0.00$ vs. $3.89 \pm 0.31$), indicating superior visual clarity, structural coherence, and reduced noise. While Tissue Visualization ($3.68 \pm 0.43$ vs. $3.86 \pm 0.30$) and Diagnostic Value ($3.68 \pm 0.55$ vs. $3.86 \pm 0.36$) scored slightly lower for Syn-Pre, the differences remain minimal and clinically acceptable. To further quantify this, one-sided Wilcoxon signed-rank tests were conducted under the null hypothesis that Syn-Pre underperforms SOC-Pre by $\geq 0.16$ points. The resulting $p$-values were 0.0159 for tissue structure and 0.0003 for

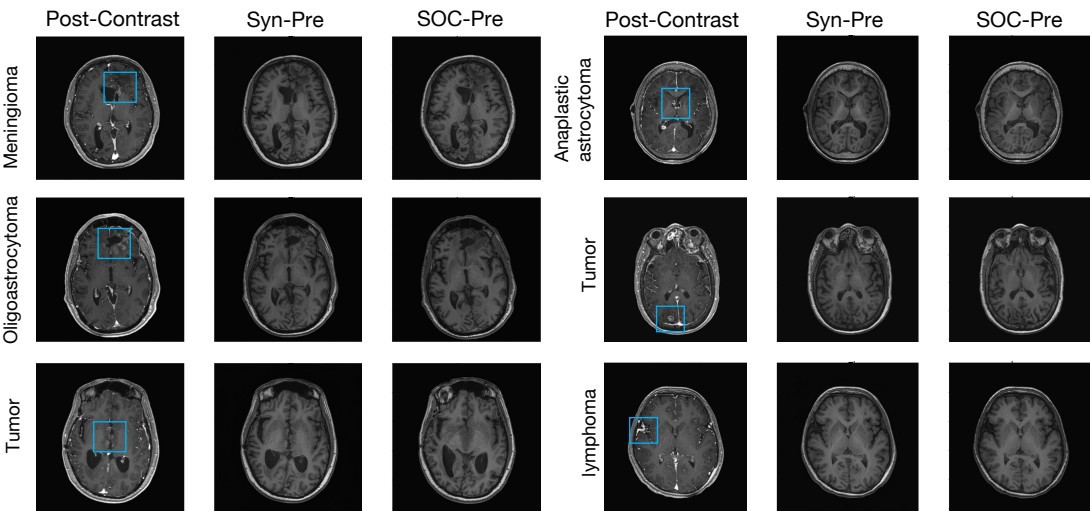

*Figure 11.* The example cases with various pathology types.

diagnostic value. Therefore, we reject this null hypothesis at the 0.05 significance level and confirm that Syn-Pre is not meaningfully worse. Together, these reader study results, combined with the extensive evaluations, reinforce the strong performance and practical viability of Syn-Pre as a reliable substitute when SOC-Pre images are unavailable or suboptimal.

# E. Discussion

## E.1. Assuming Binary Mask for Contrast Agent Concentration

In fact, assuming a uniform distribution of contrast agent uptake across specific organs simplifies the complex pharma-cokinetics of gadolinium-based contrast agents. In reality, contrast agent concentration varies spatially due to differences in vascular permeability, perfusion rates, extracellular volume fractions, and clearance mechanisms across tissues. For instance, tumors with irregular angiogenesis accumulate contrast differently than healthy tissues, while highly perfused organs such as the kidneys and liver experience rapid uptake and clearance. This variability means that a global uptake mask $M$, derived solely from contrast enhancement, may fail to fully capture the true spatial heterogeneity of [Gd], leading to potential inaccuracies in estimating pre-contrast signal.

Luckily, although the inpainting+ process was designed for structural restoration, it inherently addresses this issue by integrating information from the Post-Contrast image. Since the Post-Contrast image preserves both structural details and intensity variations related to contrast agent distribution, it provides a richer context for reconstructing pre-contrast signals. The texture patterns in enhanced regions naturally encode information about local contrast concentration, vascular properties, and tissue heterogeneity, which proves useful in refining the pre-contrast estimation.

## E.2. Base Model Investigation

This paper presents a novel physics-guided dual-stage learning framework for pre-contrast image synthesis. The proposed approach utilizes NestedUNet-based models for the autoencoder, segmentation, and inpainting+ processes. Despite employing a relatively basic convolutional architecture, the proposed SPHERE framework outperforms advanced models such as Mamba and KAN in this setting. Future work will explore alternative model architectures within this framework to further enhance performance and generalizability.

## E.3. Runtime Analysis

While the proposed model demonstrates reduced learning complexity compared to conventional approaches, it integrates three NestedUNet-based components, which increases the overall inference time. To evaluate its computational efficiency in practical applications, we conducted a runtime analysis using two standard metrics: Throughput, measured in images per second (I/s), and Latency, measured in seconds per image (s/I).

*Table 7.* Runtime analysis comparing throughput and latency across methods.

| Runtime Metric | UNet | Att-UNet | UNet++ | SwinIR | UKAN | MambaIR | BICEPS | SPHERE* | SPHERE* (FP16) |
|---|---|---|---|---|---|---|---|---|---|
| Throughput (I/s) | 1.12 | 1.06 | 0.97 | 1.04 | 1.10 | 1.33 | 1.06 | 0.45 | 0.59 |
| Latency (s/I) | 0.89 | 0.94 | 1.03 | 0.96 | 0.91 | 0.75 | 0.94 | 2.20 | 1.69 |

As shown in Tab. 7, the inference speed of the proposed SPHERE model is approximately $2\times$ slower than existing baselines, primarily due to the added complexity of the dual-stage architecture. Nevertheless, from a deployment standpoint, post-processing a full DICOM series within or around five minutes is generally acceptable in clinical workflows. With FP16 precision, SPHERE processes approximately 177 slices in under five minutes, meeting this practical constraint. If further speedup is needed, model simplification or hardware-level optimization could be considered.

### E.4. Pre-to-Post Image Synthesis Feasibility

The equation in Eq. 11 can be naturally extended to the synthesis of post-contrast MRI from pre-contrast images. However, a critical challenge remains in accurately estimating the segmentation map, which is essential for distinguishing contrast-enhanced regions. The inherent variability in contrast uptake across different tissues and imaging protocols further complicates this task, making direct mapping approaches prone to inconsistencies. Consequently, the current formulation does not fully address the complexities of pre-to-post contrast synthesis. While previous studies have investigated this problem, existing methods struggle to achieve clinically reliable results, often exhibiting artifacts or insufficient contrast enhancement. A more robust approach is needed to improve the accuracy and generalizability of Pre-to-Post synthesis.

