# OpenReview forum: "Staged and Physics-Grounded Learning Framework with Hyperintensity Prior for Pre-Contrast MRI Synthesis"
_ICML.cc/2025/Conference — ICML 2025 poster_

### Official Review · Reviewer_pkvY · 2025-03-06

**Overall Recommendation:** 2

**Summary:**

This paper is about using post-contrast MRI to create pre-contrast MRI via deep learning. Physics principles are built into the model. To tackle the complexity in setting up the model and its training, a two-stage approach is presented, which alleviates the challenge in handling the complexity. The approach is set up as an inpainting process that first learns about a mask in post-contrast MRI and then rebuilds the pre-contrast MRI.

**Claims And Evidence:**

Yes

**Essential References Not Discussed:**

No

**Experimental Designs Or Analyses:**

Yes

**Methods And Evaluation Criteria:**

Yes

**Other Comments Or Suggestions:**

Please refer to Strengths and Weaknesses.

**Other Strengths And Weaknesses:**

The strength can be found in two aspects. One is to construct pre-contrast MRI from post-contrast MRI, which is not well covered in existing research. The other is the inclusion of physics-based principles into the reconstruction model, but please see comments below.

Weaknesses are
1. Why dS_pre/dS_post appeared on both sides of Eq. (15)?
2. It is not clear how Eq. (19-20) derive the brightness prior H. By prior, it generally means the information known beforehand, for example, before MRI is acquired. But from Eq. (19-20), it seems the prior is derived from post-contrast MRI via auto-encoder and softmax, in this sense, it is unclear if that qualifies as prior information, unless I am missing something here.
Furthermore, is the prior H calculated slice-by-slice for a post-contrast MRI, or is it calculated only once for the whole post-contrast MRI series?
3. Why is there a need to use a regularization term in Eq. (22)?
4. How is \tau determined in Eq. (22)?
5. It is not clear how the deep learner in Stage 2 works or gets trained? Does it somehow incorporate the form of Eq. (11) in the deep learning model? Or is it just a deep learning model trying to mimic Eq. (11)?
6. How is the regularization term, Eq. (23), used in the loss function of Stage 1? This should be given in the main text instead of the appendix.
7. The practical applicability is a concern, as there are large number of weight parameters to tune, as given in Eq. (31) and (34). These many weight parameters may pose a difficult for users to select a good combination.
8. The relationship between Eq. (34) and Eq. (31) is confusing. It seems Eq. (34) includes the loss term given in Eq. (31), the inpainting+ loss, but according to the main text, an advantage of the paper is to separately train the masking step and inpainting step, then why does Eq. (34) involve Eq. (31)?

**Questions For Authors:**

Please refer to Strengths and Weaknesses.

**Relation To Broader Scientific Literature:**

The paper is well related to the broader scientific literature and presents a method for reconstructing a missing modality in MRI.

**Theoretical Claims:**

Yes

---

> ### Author Rebuttal · Authors · 2025-04-01
>
> Thank you for taking time and efforts to review our paper, your opinon is really appreciated. Thank you for acknowledging our work’s theoretical contribution, motivation and usage of the physcis principles. The goal of this project is to leverage the power of AI4Science to tackle unsolved challenges in MRI applications, which we believe is as equally important to achieving a higher score on well-studied tasks. For each weakness item, our response is listed as below.
>
> 1.The left is derivative, the right is partial derivative, which is minorly different.
> 2.Yes, we agree that ‘prior’ generally refers to information known beforehand, but it does not necessarily imply a temporal sequence in MRI scans. A more precise definition is: prior knowledge refers to any information about the problem beyond the training data. In our study, the prior is the assumption that contrast uptake regions typically appear hyperintense compared to non-enhanced areas [Line 201]. Equations 19–20 are designed to enforce this prior in a differential manner. The prior is preset for all training images.
> 3.Eq 22 defines the psudo ground truth for Stage 1, there is no regularization on it. But for the output of stage 1, we use regularization to ensure the model effectively learns the true hyperintensity prior like in [1]. The regularizer is assigned a low learning rate to avoid ruining the overall latent learning process.
> 4.The initial selection of τ is 0.1. This subtraction imaging technique, $\mathbf{S}{\text{post}} - \mathbf{S}{\text{pre}}$, is widely used in clinical practice [2] to capture contrast enhancement, with τ applied just to suppress background noise. Although τ is chosen empirically, its practical range is narrow (e.g., 0.08–0.12). A simple grid search on a few representative cases (e.g., τ ∈ [0.08, 0.10, 0.12]) is typically sufficient. We also explored different τ values in the ablation study (Reviewer 2, Item 1).
> 5.You are right! basically, the proposed model is trained to mimic Eq. 11. In our work, we disentangle the whole learning process to two stage learning. Different losses are designed to facilitate end-to-end model training. Similar to item 3, a light regularization term is applied to constraint the model to adapt to the physcis law.
> 6.Please refer to Item 3.
> 7.Thank you for raising this important question regarding the applicability of our model given the presence of nine weighting parameters (from Eq. 31 and 34). We offer two practical solutions:
> (1) Use of Predefined Weights:
> These weights were set based on the intrinsic properties of each loss function, not on specific data modalities or anatomies, and have yielded satisfactory results across scanners, sites, and anatomical regions. Below is the rationale for each:
> * $\lambda_{\text{L1}} = 1$: Standard for pixel-wise accuracy.
> * $\lambda_{\text{SSIM}} = 10$: SSIM ranges [0, 1], so we boost its contribution.
> * $\lambda_{\text{perceptual}} = 0.5$: Operates on high-dimensional feature space, typically yielding larger values; thus, we scale it down.
> * $\lambda_{\text{adv}} = 1$: Realism is equally important to pixel similarity.
> * $\lambda_{\text{Inpaint+}} = 1$: We treat segmentation and inpainting as equally critical.
> * $\lambda_{\text{ae}} = 1$: Autoencoder output should match L1-level fidelity.
> * $\lambda_{\text{bce}} = 10$: BCE loss yields small values over binary masks; we scale it up for balance.
> * $\lambda_{\text{hyper}} = 0.1$: Physics prior should guide but not dominate learning.
> * $\lambda_{\text{psyn}} = 0.01$: Regularization terms are generally weighted lower to avoid over-constraining.
> (2) Use of Normalized Loss Weighting:
> Alternatively, we can apply a simple approach by setting​$\lambda_{i}  = \frac{1}{\mathcal{L}_i}$ for each loss term $\mathcal{L}_i$, allowing all objectives to converge uniformly.
> These options support the generalizability and practical utility of our method without the need for extensive manual tuning.
> 8.While each stage has its own loss function, the model is trained end-to-end, similar to multi-task learning [3]. We apologize for the confusion and will clarify the separation of stage and final losses in future revisions.
>
> Thank you again for your detailed and extensive comments, we are sorry our current manuscript might cause some confusion and misunderstanding, Hope our response can address the confusions. In later revision, we will update our manscript to make it clearer according to your comments.  Feel free to let us know if there is any questions. Thank you!
>
> Reference:
> [1] Myronenko, Andriy. "3D MRI brain tumor segmentation using autoencoder regularization." Springer International Publishing, 2018.
> [2] Hubbard, C., et al. The use of MRI digital subtraction technique in the diagnosis of traumatic pancreatic injury. Radiology Case Reports, 14(5), 639-645.
> [3] Zhang, Y., & Yang, Q. (2018). An overview of multi-task learning. National Science Review, 5(1), 30-43.

---

### Official Review · Reviewer_YXmq · 2025-03-12

**Overall Recommendation:** 2

**Summary:**

This work discusses a deep learning method for recovering pre-contrasted MRI from post-contrasted MRI. The authors propose to first estimate a thresholded map as a mask indicating contrast agent update. This mask is then fed as an additional conditioning signal for recovering the pre-contrast image. The authors also discussed an artifact removal approach for enhancing pre-contrast images. The proposed framework is evaluated against several other types of basic feed forward neural network architectures.

## update after rebuttal

I would like to thank the authors for the efforts especially for the additional comparisons and clarifications.

With that being said, I still find the manuscript suffers somehow from unjustified claims / design choices and flaws in writing structures that are very unlikely to be addressed in a revision (and it may not be of the best practice to first introducing two heavy components: AE and a complicated artefact removal framework which had led to quite a lot confusions and then suddenly claiming they are not essential or are just for extension purpose). Also, the huge amount of hyper-parameters (as pointed out by Reviewer `pkvY`) render the proposed framework very difficult to be applied to other datasets by readers. I would therefore keep my current rating.

**Claims And Evidence:**

Claim: Line 095 left: The proposed work "present a significant advancement in MRI image by developing a method capable of generating high-quality pre-contrast images"
Fact check: This claim cannot be substantiated unless being compared with commonly used medical image translation / quality enhancement methods such as [1-6]. Given the similar mathematical formulation (pixel to pixel mapping), even some of them are originally designed for slightly different purpose, re-purposing them to the pre-contrast recovery should be straightforward.

Claim: Line 058 right: The proposed methods are extensively evaluated on two real-world datasets we collected from two hospitals, demonstrating their robustness.
Fact check: Robustness are normally characterized by stable performance against OOD data or adversarial samples at test time. Unlikely to be the case described in the Experiment section.

Claim: Training a simple image-to-image synthesis network to map Post-Contrast to Pre-Contrast images often fails to balance the synthesis of the structural and contrast information in the image.
Fact check: Line 158 left: M is still a function of S_{post} though.

Claim: Line 195 left: "...a conventional deep segmentation model to guarantee model precision and robustness ..."
Fact check: convolutional segmentation models alone cannot guarantee model precision and robustness without proper training data / training approach.

Claim: Eq. 16 The proposed method reduced the complexity from multiplicative to additive.
Fact check: It is unclear how Eq. 16 is reached given the sequential nature of Eq. 11 and 12.

Claim Line 265 right: "these methods represent the golden standard approaches in image enhancement, segmentation, and synthesis."
Fact check: No evidence. E.g. for image enhancement and synthesis there exists much more advanced methods such as [1-6].

1. Adaptive latent diffusion model for 3d medical image to image translation: Multi-modal magnetic resonance imaging study
2. Unsupervised Medical Image Translation With Adversarial Diffusion Models
3. DuDoDR-Net: Dual-domain data consistent recurrent network for simultaneous sparse view and metal artifact reduction in computed tomography
4. Target-guided diffusion models for unpaired cross-modality medical image translation
5. Cascaded multi-path shortcut diffusion model for medical image translation
6. A generic deep learning model for reduced gadolinium dose in contrast‐enhanced brain MRI

**Essential References Not Discussed:**

Works on medical image enhancement / modality translation / artifact removal, such as  [1-6] should be discussed, given the similar mathematical formulations (learning pixel to pixel mappings in image intensity domain).

**Ethical Review Concerns:**

N/A, given the claims in line 236-237 right.

**Experimental Designs Or Analyses:**

The claimed superiority cannot be substantiated without systematic comparisons with recent medical image enhancement / modality translation / artifact removal works, such as  [1-6]. Given the similar mathematical formulation (pixel to pixel mapping), even some of them are originally designed for slightly different purposes, applying them to pre-contrast recovery should be straightforward.

**Methods And Evaluation Criteria:**

The evaluation metrics are relevant. However, given that most experiments are performed on two in-house datasets, the reproducibility of the proposed work is unclear.

**Other Comments Or Suggestions:**

N/A

**Other Strengths And Weaknesses:**

Strengths:
- The authors have presented the mathematical model for image contrast enhancement.
- Improved performances are shown compared with experimented neural networks.

Weaknesses:
- The paper suffer from a lack of rationale for many design choices: E.g., Why do the authors process S_{post} with an autoencoder?
Sec. 2.3 can be distracting as it is not closely centered at the pre-contrast recovery problem.
- The motivation for processing S_{post} with AE is unclear.
- The threshold \tau is subject to manual choice and it is critical for defining M_{true}. Given the heterogeneity in real-world MRI acquisition and its non-quantitative nature choosing a proper \tau can be difficult in real world.
- Artifact removal: It is related but not centered on the pre-contrast recovery. It should instead be put as a standalone work and carefully assessed alone. Also, little information about the rationale behind and the methodology is presented in the main text.

**Questions For Authors:**

Given that most experiments are performed on two in-house datasets, how would the reproducibility of the proposed work be assessed?

**Relation To Broader Scientific Literature:**

This work falls into the categories of medical image quality enhancement / modality translation / artifact removal, given the similar mathematical formulations.

**Theoretical Claims:**

The proposed work is mostly empirical. Please find my additional comments in `Claims And Evidence` section.

---

> ### Author Rebuttal · Authors · 2025-04-01
>
> Thank you for your in-depth review of our manuscript. Your comments are constructive and will help improve the quality of our work. In this manuscript, we propose a novel MRI theory-driven method to address a challenging problem in MRI imaging. Please find our detailed responses to your comments below:
>
> Claim:
> 1.(Comparison to other methods such as {1-6}): Thank you for raising this point. Per your suggestion, we conducted three additional comparative experiments using models No. 2, 3, and 6 from your recommended list. For No. 3 and 6, we adapted them to the same pipeline used by SPHERE. For the Syn-Diff model (No. 2), we trained it for 40 epochs, which took approximately 80 hours on two A100 GPUs. The quantitative results are summarized as follows:
> Tab 5. Additional Comparasions
> | Models | PSNR ↑   | SSIM ↑  | CNR ↓   | LPIPS ↓ | GMSD ↓  | CFS ↑   | IRC ↑   | EIS ↑   |
> |------------------|---------|--------|--------|--------|--------|--------|--------|--------|
> | DuDoDR-Net     | 32.6218 | 0.7974 | 0.1078 | 0.0738 | 0.0404 | 0.6854 | 0.8143 | 0.2222 |
> | Dose Reduction | 34.8808 | 0.8665 | 0.0804 | 0.0492 | 0.0358 | 0.8450 | 0.7847 | 0.3941 |
> | Syn-Diff	            | 35.7180 | 0.8961 | 0.0728 | 0.0337 | 0.0325 | 0.7068 | 0.8633 | 0.5434 |
> | SPHERE*           | 36.8244 | 0.9026 | 0.0628 | 0.0313 | 0.0315 | 0.9181 | 0.8684 | 0.5364 |
>
> Please also refer to Fig 3. in https://anonymous.4open.science/r/ICML25_rebuttal-63F3 for qualitative evaluation. DuDoDR-NET and Dose Reduction perform significantly worse, while Syn-Diff indicates somewhat closer results. However, it overemphasizes background regions and fails on pathological structures. These outcomes further support our claim that existing models struggle to balance structural fidelity and contrast enhancement. In contrast, our model provides the most effective solution to this challenging task.
> 2.(Line 058 right): We totally agree with you on this, Sorry for this mistake.
> 3.(M is still a function of S_{post}), we will add one more equation of M in terms of S_{post} to make it clearer.
> 4.(Line 195 left): Yes, the output of the arbitrary segmentation model is combined with the Hyperintensity branch to make a fused prediction, the loss function is applied on the fused prediction for training. We will refine the wording in revision.
> 5.(Unclear how Eq. 16 is reached): Yes, as noted, our task inherently involves a sequential combination of segmentation and inpainting+ subtasks. With our dual stage learning, the complexity is additive with modular training [1]. However, directly modeling these together with a single network needs to encode all segmentation-to-inpainting mappings simultaneously. This multiplicative complexity arises naturally from the fact that the model needs encode every possible combination of segmentation-to-inpainting mappings. We will explicitly clarify this rationale in our revision.
> 6.[Evaluation in-house datasets] In this study, we tried our best to evaluate our model more extensively like across multiple datasets, sites, anatomies, and downstream tasks. But I am sorry we cannot disclose datasets due to license constrait. In future, we plan to apply our model to public datasets to further support reproducibility.
>
> Weakness:
> 1,2. The autoencoder is a marginal component of our model. It works with a scaling factor to adjust image intensity, which mgiht also be handled by raw input. We keep it to subtly suppress non-structural noise, making hyperintensity extraction more robust. Though not explicitly designed for denoising, the AE learns compressed bottleneck representations that capture structural and semantic manifold, naturally reducing inconsistent noise or redundancy [2]. We apologize for missing this rationale and will include it in the next revision. An ablation study on the AE is also provided in Fig 4. in Link for details. Overall, we find AE marginally beneficial.
> 3.Please refer to Reviewer #4 Item 4.
> 4.The artifact removal component is included to extend our model’s applicability to scenarios with corrupted images. The current implementation serves as a proof of concept, and we plan to explore it further in future work.
>
> Thank you again for taking the time to rigorously review our paper! Your comments will undoubtedly enhance the overall quality of our work like on the model rationale. We sincerely appreciate your thoughtful feedback. Please feel free to reach out if you have any further questions.
>
> Reference
> [1] Leung, K. H., et al (2020). A physics-guided modular deep-learning based automated framework for tumor segmentation in PET. Physics in Medicine & Biology, 65(24), 245032.
> [2] Bartlett, O. J., et. al (2023). Noise reduction in single-shot images using an auto-encoder. The Royal Astronomical Society, 521(4), 6318-6329.

---

### Official Review · Reviewer_14Xf · 2025-03-13

**Overall Recommendation:** 3

**Summary:**

This paper proposes SPHERE, a staged and physics-grounded learning framework for synthesizing Pre-Contrast MRI images from Post-Contrast MRI scans. The key innovation lies in incorporating MRI physics principles and a hyperintensity prior into a two-stage deep learning model. The framework consists of segmentation and inpainting. Extensive experiments on multi-site MRI datasets demonstrate that SPHERE outperforms existing deep learning methods across multiple metrics and generalizes well to spine and breast MRI applications. The method also extends to artifact removal for corrupted Pre-Contrast images. The approach has potential clinical significance by reducing the need for additional imaging sessions, cost, and patient risk.

**Claims And Evidence:**

Claim 1: SPHERE synthesizes clinically viable Pre-Contrast MRI images from Post-Contrast scans.
Evidence:
The method is tested on two large, real-world MRI datasets from multiple sites and scanners. It achieves higher PSNR, SSIM, and CNR compared to baseline models, supporting the claim of high-fidelity image synthesis.

Claim 2: The two-stage learning framework improves synthesis quality over direct image-to-image translation.
Evidence:
The paper provides a mathematical derivation of the complexity reduction and gradient stability benefits of the two-stage approach. Empirical results show that SPHERE outperforms state-of-the-art methods, which struggle with contrast preservation and structural accuracy.

Claim 3:
The hyperintensity prior improves contrast segmentation and Pre-Contrast reconstruction.
Evidence: The segmentation results demonstrate that incorporating a hyperintensity prior enhances contrast region detection. Comparative experiments indicate improved structural and contrast preservation.

Claim 4: SPHERE generalizes to other medical imaging tasks.
Evidence:
The model is fine-tuned on spine and breast MRI datasets, achieving strong quantitative results, demonstrating adaptability beyond brain MRI.

Weaknesses in evidence:
Weakness 1: The effectiveness of the hyperintensity prior is mentioned, but an explicit ablation study isolating its impact is missing.
Weakness 2: While the results indicate strong performance, validation with radiologists or clinical usability studies would further substantiate the claim of clinical applicability.

**Essential References Not Discussed:**

No

**Experimental Designs Or Analyses:**

Yes

**Methods And Evaluation Criteria:**

The proposed MRI physics guided SPHERE framework is well-aligned with the problem of Pre-Contrast MRI synthesis. The two-stage learning approach, incorporating a hyperintensity prior, is a well-motivated methodological choice. The formulation effectively addresses the limitations of direct image-to-image translation by improving contrast segmentation and synthesis accuracy.
The evaluation criteria are appropriate for assessing image synthesis quality. The authors use standard image quality metrics, including PSNR, SSIM, CNR, and LPIPS, which are widely accepted for medical image analysis. The inclusion of multi-site, multi-scanner datasets enhances the robustness and generalizability of the findings. Additionally, downstream tasks (e.g., low-dose contrast simulation, spine and breast MRI applications) provide further validation of clinical utility.

**Other Comments Or Suggestions:**

No

**Other Strengths And Weaknesses:**

Strengths
1. The paper presents a novel two-stage physics-grounded approach for Pre-Contrast MRI synthesis, integrating MRI signal modeling with deep learning, which is an innovative contribution beyond purely data-driven synthesis methods.

2. The method addresses a practical problem in medical imaging, reducing the need for additional scans, which could lead to cost savings and reduced patient risk. The evaluation on real-world multi-site datasets enhances its potential clinical impact.

3. The experiments are rigorous and diverse, including:
Comparisons with strong baselines (UNet, SwinIR, UKAN).
Multiple quantitative metrics (PSNR, SSIM, CNR, LPIPS).
Downstream clinical applications (spine, breast MRI, and low-dose contrast simulation).

4. The paper is generally well-structured with detailed methodological explanations.

Weaknesses
1. While the paper presents intuitive justifications for the hyperintensity prior and two-stage framework, an explicit ablation study quantifying their contributions is missing.

2. The two-stage design is claimed to be computationally more efficient, but there are no runtime comparisons or training time analyses to support this claim.

3. While the method performs well quantitatively, there is no validation by radiologists to confirm the clinical realism and usability of the synthesized Pre-Contrast images.

**Questions For Authors:**

No

**Relation To Broader Scientific Literature:**

This paper builds on existing work in medical image synthesis, MRI reconstruction, and physics-informed deep learning while introducing a novel two-stage, physics-grounded approach for Pre-Contrast MRI synthesis.

Medical image synthesis:
Prior works, such as UNet-based image-to-image translation and transformer-based synthesis models, have been applied to MRI reconstruction but often fail to preserve contrast details when synthesizing missing sequences.
The proposed SPHERE framework extends these efforts by explicitly modeling MRI physics to improve synthesis quality.

Physics-Guided deep learning in MRI:
The paper aligns with trends in physics-informed learning. Unlike purely data-driven approaches, SPHERE incorporates MRI signal equations to constrain the learning process, similar to prior work in quantitative MRI reconstruction.

**Theoretical Claims:**

Yes

---

> ### Author Rebuttal · Authors · 2025-04-01
>
> Thank you for your meticulous and comprehensive review. We appreciate your recognition of our work on quantitative performance, mathematical support, prior knowledge incorporation, and model generalizability. Regarding the identified weaknesses, we provide the following responses:
> 1.Yes, the hyperintensity prior and two-stage framework are two critical components of our model, and quantifying their contribution to performance is important. We have added ablation studies on key components such as the Hyperintensity Prior, the Arbitrary Segmentation Branch, the Autoencoder module, the Dual Learning stage, and selection of τ.
> Tab 3. Ablation study on different modules and τ selection.
> | Configuration               | PSNR ↑   | SSIM ↑  | CNR ↓   | LPIPS ↓ | GMSD ↓  | CFS ↑   | IRC ↑   | EIS ↑   |
> |----------------------------|---------|--------|--------|--------|--------|--------|--------|--------|
> | W/o Hyperintensity          | 35.7608 | 0.8842 | 0.0729 | 0.0434 | 0.0356 | 0.8970 | 0.8252 | 0.4041 |
> | W/o Abitrary Seg             | 36.3764 | 0.8951 | 0.0666 | 0.0346 | 0.0337 | 0.9068 | 0.8558 | 0.4703 |
> | W/o AE                    | 36.6764 | 0.9006 | 0.0635 | 0.0321 | 0.0328 | 0.9119 | 0.8585 | 0.5075 |
> | W/o Dual Step      | 34.5522 | 0.8580 | 0.0836 | 0.0479 | 0.0362 | 0.8462 | 0.8454 | 0.3746 |
> | SPHERE*           | 36.8244 | 0.9026 | 0.0628 | 0.0313   | 0.0315  | 0.9181 | 0.8684 | 0.5364 |
> | **τ selection** |   |	|	|	|	|	|	|	|
> | τ = 0.06                   | 35.7835 | 0.8990 | 0.0651 | 0.0366 | 0.0320 | 0.9109 | 0.8560 | 0.5366 |
> | τ = 0.08                   | 36.8841 | 0.9034 | 0.0628 | 0.0298 | 0.0315 | 0.9179 | 0.8676 | 0.5319 |
> | τ = 0.10                   | 36.8244 | 0.9026 | 0.0628 | 0.0313 | 0.0315 | 0.9181 | 0.8684 | 0.5364 |
> | τ = 0.12                   | 36.8759 | 0.9035 | 0.0624 | 0.0299 | 0.0317 | 0.9176 | 0.8687 | 0.5333 |
> | τ = 0.14                   | 36.7256 | 0.9008 | 0.0635 | 0.0313 | 0.0320 | 0.9097 | 0.8702 | 0.5246 |
>
> Above quantitative and qualitative results in Fig. 4 on https://anonymous.4open.science/r/ICML25_rebuttal-63F3 shows that the key component such as Dual stage learning, Hyperintensity, and arbitrary seg are all beneficial to the model performances to different extents. For the τ selection, values in the range of 0.08–0.14 generally yield consistent performance with minimal variation. This suggests that our model is not highly sensitive to exact choice of τ, indicating robustness and fewer constraints on hyperparameter tuning.
>
> 2.We actually do not intend to claim the computational efficency, Instead, we aim to highlight that the learning complexity or difficulty of the dual-stage learning method is lower than that of direct learning, as shown in Eq. 17. This does not necessarily imply faster model runtime. We apologize for any misunderstanding this may have caused. In terms of inference time, we added an runtime analysis per your request. Two metrics including throughput and latency are employed to measure the model runtime. Results are shown as below:
> Tab 4. Runtime Analysis
> | Runtime Metric       | UNet | Att-UNet | UNet++ | SwinIR | UKAN | MambaIR | BICEPS | SPHERE* | SPHERE* (FP16) |
> |----------------------|------|----------|--------|--------|------|---------|--------|---------|----------------|
> | Throughput (I/s)     | 1.12 | 1.06     | 0.97   | 1.04   | 1.10 | 1.33    | 1.06   | 0.45    | 0.59           |
> | Latency (s/I)        | 0.89 | 0.94     | 1.03   | 0.96   | 0.91 | 0.75    | 0.94   | 2.20    | 1.69           |
>
> As shown in the table, the model speed of the proposed method is approximately 2× slower than other methods due to the dual-stage design. From an application perspective, we consider post-processing of a DICOM series within or ~ 5 minutes to be clinically viable. The current runtime of SPHERE (FP16) roughly meets this criterion (~177 slices in 5m). If further acceleration is needed on high resolution 3D scan, we can refer to TensorRT or Triton for faster inference on deployment. For training on a single A100 GPU, the time required for a direct learning model such as UNet++ is 2.02h, while SPHERE requires 6.10h due to its dual-stage optimization. We have also benchmarked the training time on different hardware platforms including V100, A100, and H100. Our latest setup enables training completion in 3.1 hours, significantly accelerating model development. Please refer to Fig. 5 in the external link for more details. So, we can conclude that our dual design may need more runtime for training/inference, but within the clinically acceptable range. Thank you for raising this valuable point. We will include it in the main text in a later revision.
>
> 3. Please refer to Reviewer #1 item 1, thanks.
>
> Thank you again for your careful review of our paper, your acknowledegement of our work is really inspiring to us, and we hope we addressed your concerns, if there is any other questions, feel free to let us know! Thank you!

---

### Official Review · Reviewer_q8jE · 2025-03-13

**Overall Recommendation:** 3

**Summary:**

This paper proposes a novel staged, physics-grounded learning framework with a hyperintensity prior to synthesize Pre-Contrast images directly from Post-Contrast MRIs. The proposed method can generate high-quality Pre-Contrast images, thus, enabling comprehensive diagnostics while reducing the need for additional imaging sessions, costs, and patient risks. The authors claim it is the first Pre-Contrast synthesis model capable of generating images that may be interchangeably used with standard-of-care Pre-Contrast images. Extensive evaluations across multiple datasets, sites, anatomies, and downstream tasks demonstrate the model’s robustness and clinical applicability, positioning it as a valuable tool for contrast-enhanced MRI workflows.

## Update after rebuttal:

I appreciate the authors’ responses to my questions, especially involving experts to assess the quality of the synthesized pre-contrast images. I have updated my score accordingly.

However, I'd like to point out that if you use a t-test to check the significance of the 'Reader Scores', you will find no significant differences between them, for example, 4.00 ± 0.00 vs. 3.89 ± 0.31. You might need to reformulate your words instead of claiming 'consistently outperform', which doesn't impact your conclusion, though.

**Claims And Evidence:**

Yes

**Essential References Not Discussed:**

N/A

**Experimental Designs Or Analyses:**

Yes. The whole experimental sections.

**Methods And Evaluation Criteria:**

Yes

**Other Comments Or Suggestions:**

N/A

**Other Strengths And Weaknesses:**

Strengths:

1. The two-stage training paradigm (initial physics-based augmentation followed by deep-learning-based synthesis) seems innovative.
2. The proposed model achieves better results (quantitatively and qualitatively) compared to prior work, indicating the success of integrating domain knowledge into the learning process.
3. The proposed approach has the potential to reduce the need for multiple MRI scans while maintaining diagnostic quality.

Weaknesses:

1. It is unclear how well the generated synthetic pre-contrast images preserve pathology-related features in conditions like tumors, multiple sclerosis, or stroke. I am expecting to see more in-depth clinical validations as this paper focuses on real-world problems
2. How can the proposed method be generalized to different MRI sequences and modalities?
3. I am also wondering how each component of the proposed method benefic the performances. Could you please provide a more detailed ablation study to explicitly evaluate the impact of individual components?

**Questions For Authors:**

N/A

**Relation To Broader Scientific Literature:**

The paper can be used to deal with the real-world MRI generation problem.

**Theoretical Claims:**

N/A

---

> ### Author Rebuttal · Authors · 2025-04-01
>
> Thank you for reviewing our manuscript from the clinical application perspective, your comments are truly constructive for us. We appreciate your recognition of our novelty, integration of domain knowledge, and potential clinical applicability. The motivation of our study is to provide a theoretically supported AI solution to address an unsolved application problem in MR imaging. Regarding the weakness, we provide the responses as below: (an External Link for Results Visualization: https://anonymous.4open.science/r/ICML25_rebuttal-63F3)
> 1.Thank you for raising this concern about clinical validation. We have invited two independent radiologists to comprehensively assess the quality of the synthesized pre-contrast images in comparison to the SOC pre-contrast images. Reader #1 has over 15 years of clinical experience in radiology, and Reader #2 has over 10 years. A total of 15 cases were reviewed. The pathologies included tumors, GBM (glioblastoma), lymphoma (CNS and Hodgkin’s-related), anaplastic astrocytoma, meningioma, oligoastrocytoma, Von Hippel-Lindau disease, and fungal/parasitic infection. Five metrics were used: Perceived Image Quality, Anatomical Alignment, Tissue Visualization (Usability with Post), Diagnostic Value (when +Post), and Imaging Artifacts. All metrics were scored using a 1–4 Likert scale. Please refer to Table 1 in the link for more metric details. The results from both readers are summarized as follows (see Fig. 1 and Fig. 2 in the link for additional statistical analysis and visualized results):
> Tab 2. Reader Scores
> | Metric                         | Syn-Pre Mean ± SD, | Syn-Pre Quartiles, | SOC-Pre Mean ± SD, | SOC-Pre Quartiles |
> |-------------------------------|-------------------|-------------------|-------------------|-------------------|
> | Perceived Image Quality       | 3.96 ± 0.19       | 4.00, 4.00        | 3.64 ± 0.49       | 3.00, 4.00        |
> | Anatomical alignment | 3.96 ± 0.19       | 4.00, 4.00        | 3.75 ± 0.52       | 4.00, 4.00        |
> | Tissure Visualization       | 3.68 ± 0.43       | 3.50, 4.00        | 3.86 ± 0.30       | 4.00, 4.00        |
> | Diagnosis Value   | 3.68 ± 0.55       | 3.00, 4.00        | 3.86 ± 0.36       | 4.00, 4.00        |
> | Imaging Artifacts             | 4.00 ± 0.00       | 4.00, 4.00        | 3.89 ± 0.31       | 4.00, 4.00        |
>
> As demonstrated by the results, the Syn-Pre images consistently perform better or comparable to SOC-Pre in several important aspects. Specifically, Syn-Pre achieved higher scores in perceived image quality (3.96 ± 0.19 vs. 3.64 ± 0.49), anatomical alignment with post-contrast (3.96 ± 0.19 vs. 3.75 ± 0.52), and imaging artifacts (4.00 ± 0.00 vs. 3.89 ± 0.31), suggesting superior visual clarity, structural coherence, and reduced noise. While visualization of tissue (3.68 ± 0.43 vs. 3.86 ± 0.30) and diagnostic value when paired with post-contrast (3.68 ± 0.55 vs. 3.86 ± 0.36) scored slightly lower for Syn-Pre, the difference remains minimal and clinically acceptable. Specifically, 12 out of 15 cases were rated equivalent to SOC-Pre in these two metrics, with only 3 cases showing slight degradation. To further quantify this, one-sided Wilcoxon signed-rank tests were conducted under the null hypothesis that Syn-Pre underperforms SOC-Pre by ≥0.16 points. The resulting P-values are 0.0159 for tissue structure and 0.0003 for diagnostic value. Therefore, we reject this null hypothesis and confirm that Syn-Pre is not meaningfully worse. Together, these reader study results together with the extensive evaluations reinforce the strong performance and the practical viability of Syn-Pre as a reliable substitute when SOC-Pre images are unavailable or suboptimal.
>
> 2.For the common MRI sequences or modalities such as T1, T2, T2FLAIR, T2STAR, TOF, TRICKS, ADC, ASL, DWI, LOC, SSFP, and SWI, gadolinium-based contrast agents (GBCAs) are predominantly used in T1-weighted imaging, which serves as the clinical standard for contrast enhancement [1]. We acknowledge that contrast agents have limited but notable applications in other sequences such as T2-STIR, TOF, and SWI. The fundamental prerequisite for applying our method is the presence of hyperintense in the post-contrast images, which are not visible in the pre-contrast images. When this condition is met, our dual-stage learning framework which is driven by hyperintensity priors, can theoretically be extended to other sequences beyond T1. In future work, we plan to explore the generalizability of our approach to additional modalities.
>
> 3.Please refer to Reviewer #2 item 1.
> Thank you again for taking time and efforts to review our paper, please feel free to raise a question if there is any confusion. Thank you!
> Reference
> [1] Lohrke, Jessica, et al. "25 years of contrast-enhanced MRI: developments, current challenges and future perspectives." Advances in therapy 33 (2016): 1-28.

---

### Decision · Program_Chairs · 2025-05-01

**Decision:**

Accept (poster)

**Comment:**

This manuscript presents SPHERE, a staged physics-grounded learning framework for synthesizing pre-contrast MRI from post-contrast scans. The approach integrates MRI physics principles and a hyperintensity prior into a two-stage deep learning model comprising segmentation and inpainting steps. The reviewers acknowledged the paper's novelty in addressing an important clinical challenge and its thorough theoretical foundation. After rebuttal, the authors provided extensive additional experiments including reader studies with radiologists, ablation analyses, and comparisons with recent methods, which strengthened the empirical validation. The clinical reader study demonstrated that synthetic pre-contrast images achieved comparable or superior scores to standard-of-care images across multiple quality metrics. However, some reviewers remained concerned about aspects of the methodology, particularly the large number of hyperparameters and clarity of certain design choices. While the authors clarified that the autoencoder and artifact removal components were optional extensions rather than core elements, there were differing views on whether the manuscript structure effectively conveyed this. The reviewers' scores showed limited convergence after rebuttal, with two maintaining weak accept recommendations and two keeping weak reject positions, primarily due to presentation rather than fundamental technical issues. The extensive experimental validation and clinical relevance were viewed positively, though opportunities remain to improve the clarity of the methodological exposition.